# AdaMeZO: Adam-style Zeroth-Order Optimizer for LLM Fine-tuning Without Maintaining the Moments

## Abstract

Fine-tuning LLMs is necessary for dedicated downstream uses, but classic back-propagation approaches necessitate a large amount of GPU memory. To this end, a recent work, MeZO, which relies solely on forward passes to fine-tune LLMs, significantly reduces GPU requirements at the cost of slower convergence due to its indifference to loss landscapes. Standard solutions, such as Adam, explore loss landscapes by estimating the first and second-order moments and storing them in memory to guide the models in moving faster through dimensions with smaller curvature and vice versa. However, directly applying Adam negates MeZO's advantage as it will triple the memory requirement. In light of this, we propose AdaMeZO, a zeroth-order optimizer enhanced by Adam-style first and second moments estimates, but without maintaining them in memory. We present a theoretical analysis of AdaMeZO, corroborated by extensive experiments demonstrating AdaMeZO's performance, showing that AdaMeZO can outperform MeZO while requiring up to 70% fewer forward passes. Visualizations of trajectories on toy functions to affirm AdaMeZO's ability to adapt to different loss landscapes. Codes are available at `https://anonymous.4open.science/r/AdaMeZO-4547/`.

## 1 Introduction

Table 1: Key features for AdaMeZO and methods in comparison. $P$ in the first column denotes the amount of memory required to store the model weight, and $B \gg P$ denotes the amount of memory required to perform backpropagation. $\delta \ll 1$ is a small positive number. "FP" abbreviates forward pass.

| | Param. memory | FP per step | 1st moment | 2nd moment |
|---|---|---|---|---|
| Adam Kingma & Ba (2014) | $3P + B$ | 1 | ✓ | ✓ |
| MeZO Malladi et al. (2023) | $P$ | 2 | ✗ | ✗ |
| HELENE Zhao et al. (2024a) | $3P$ | 2 | ✓ | ✓ |
| HiZOO Zhao et al. (2024b) | $2P$ | 3 | ✗ | ✓ |
| AdaMeZO | $(\mathbf{1} + \delta)\mathbf{P}$ | $\mathbf{2}$ | ✓ | ✓ |

Fine-tuning LLMs is necessary for dedicated downstream uses and has gained significant attention recently. Many works have emerged that aim to tune models while accessing as little memory as possible. Popular first-order methods known as parameter-efficient fine-tuning (PEFT) to alleviate the heavy memory cost by modifying only a small (potentially extra) part of the whole model Hu et al. (2022); Li & Liang (2021); Lester et al. (2021); Dettmers et al. (2023); Pan et al. (2024). Additionally, a zeroth-order method Malladi et al. (2023) implies the possibility of discarding back-propagation, the primary memory cost contributor in LLM fine-tuning, making it accessible for resource-limited devices.

As shown in Table 1, MeZO features an SGD-styled Rumelhart et al. (1986); Bottou et al. (2018) update rule, allowing in-place parameter modification. After in-place model perturbation for gradient projection estimation, the gradients are not dumped into memory but generated by a pseudo-

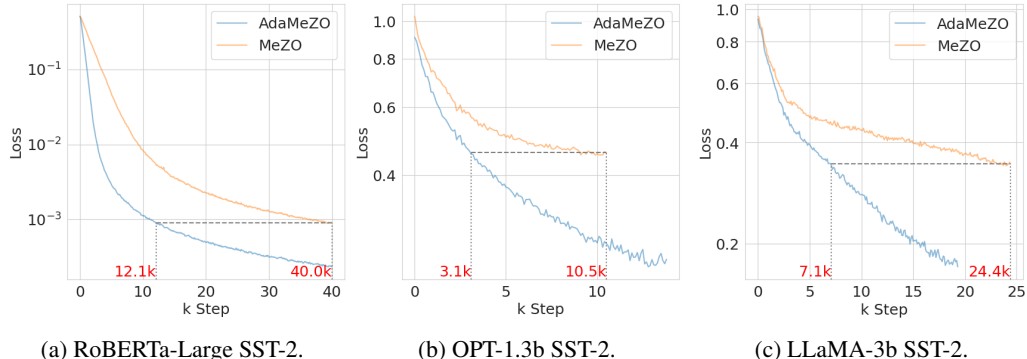

(a) RoBERTa-Large SST-2.  (b) OPT-1.3b SST-2.  (c) LLaMA-3b SST-2.

Figure 1: Loss curves of MeZO and AdaMeZO on the SST2 task. When fine-tuning RoBERTa-large, OPT-1.3b, LLaMA-3b, AdaMeZO took $69.75\%, 70.48\%, 70.90\%$ fewer forward passes to reach the loss values of MeZO at terminations, respectively. Hyperparameters and terminal conditions are detailed in Section B.3.

random number generator (PRNG) before being scaled by the previously computed projection, so the memory cost for fine-tuning is reduced to the equivalent of deploying one. However, updating the model with only the most recent gradient estimation can result in worse convergence, especially with noisy and isotropic zeroth-order gradient estimations. In comparison, adaptive optimizers like Adam Kingma & Ba (2014) and AdamW Loshchilov & Hutter (2017) that correct updates with preconditioners are more widely adopted options since the loss landscapes of LLMs exhibit complicated curvature spectra across different dimensions as documented in Sagun et al. (2016); Ghorbani et al. (2019); Zhang et al. (2023); Das et al. (2024).

However, adaptive optimizers keep historical gradient information in memory. In the case of Adam, they are the first and second moments, which are the accumulation of gradients and quadratic gradients. In other words, two vectors of the same size as the model need to be kept in memory. Considering that first-order methods use backpropagation, the additional memory cost is relatively small. But in the context of zeroth-order optimizers, the memory cost is multiplied.

Adaptive zeroth-order optimizer for LLM fine-tuning has gained recent research interest, as shown in Table 1. Pioneering works include HiZOO Zhao et al. (2024b), ZO-AdaMU Jiang et al. (2024), and Helene Zhao et al. (2024a). HiZOO proposes approximating the diagonal Hessian with an additional forward pass oracle, which doubles the memory requirement for storing the diagonal Hessian. Helene is a more direct integration of zeroth-order gradient estimation and an Adam optimizer, and ZO-AdaMU replaces the moments with an uncertain version. As a result, the memory requirement is tripled to store both diagonal Hessian estimation and cumulative history gradients. However, despite the substantial increase in memory cost, they still use a much smaller memory than first-order approaches and exhibit a noticeable performance gain compared to MeZO.

In light of the above, we introduce AdaMeZO, a zeroth-order optimizer that utilizes Adam-style first and second moments to enhance convergence without requiring additional memory to store them. This is made possible by 1) computing truncated moments that discard outdated gradients rather than faithfully maintaining the full moment estimations, and 2) block-wise generation of random gradient direction with a finer operation of the PRNG. As a result, AdaMeZO significantly reduces the number of forward passes required for convergence and improves the fine-tuned model's performance. A summary of the contributions of this work is as follows.

1. We introduce AdaMeZO, an optimizer that runs on zeroth-order gradient estimations and updates with Adam-style first and second moments. Although the moments are necessary to compute the model updates, with truncated approximations and finer operations of the PRNG, they do not need to be stored in memory. In this way, AdaMeZO can theoretically use no additional memory to improve convergence with preconditioning.

2. We establish a convergence bound of AdaMeZO under a non-convex assumption that recovers the convergence rate of preconditioned MeZO with multiples of memory cost.

3. We conduct extensive experiments to evaluate AdaMeZO's performance of AdaMeZO. We first employ 2-dimensional toy functions and visualize the optimizing trajectories. They demonstrate that AdaMeZO can converge to optimal points, whereas MeZO cannot under the same step budgets. Then we demonstrate AdaMeZO's performance by fine-tuning different models (RoBERTa Liu et al. (2019b), OPT Zhang et al. (2022a), and LLaMa Touvron et al. (2023)) for a task set identical to MeZO's. It is found that AdaMeZO almost always reaches an identical termination condition compared to MeZO, with up to 70% fewer forward passes and at higher performance.

## 2 Related Works

### 2.1 Zeroth-order Optimizers for LLMs

Zeroth-order optimization is also known as derivative-free or black-box optimization. Priorly, it is used for circumstances where objective functions have no derivatives or when obtaining the derivatives is expensive. Fine-tuning LLMs falls into the latter case and sometimes both for non-differentiable objectives Tang et al. (2023); Zhang et al. (2024a). In the context of modern deep learning, it translates to the emission of auto-differentiation by backward propagation Rumelhart et al. (1986), resulting in hugely reduced memory consumption. Some past work on zeroth-order optimizers include Spall (1992; 1997); Vakhitov et al. (2009); Agarwal et al. (2009); Raginsky & Rakhlin (2011); Jamieson et al. (2012); Wang et al. (2020); Baines et al. (2021). MeZO Malladi et al. (2023) firstly adopts the classical SPSA Spall (1992) to fine-tune billion-level dimension LLMs based on low rank assumptions on LLM fine-tuning, achieving comparable performance with much fewer GPU hours. A survey on concurrent extensions on top of MeZO can be found in Section A. Notably, from-scratch zeroth-order optimization on smaller networks is also of great research interests Chen et al. (2023).

### 2.2 First-order Optimizers for LLMs

First-order optimization algorithms form the backbone of training or fine-tuning LLMs, offering computational efficiency and scalability across billions of parameters. One of the most classic solutions is Adam Kingma & Ba (2014), featuring first and second moments corrected updates. Some of its variants are as follows. AdamW Loshchilov & Hutter (2017) introduces adaptive learning rates via moment estimates, achieving faster convergence on nonconvex objectives. LAMB You et al. (2019) features a layer-wise adaptation strategy to accelerate the training of large models employing large batches. Adafactor Shazeer & Stern (2018) reduces memory usage by maintaining factored second-moment estimates rather than the faithful estimates. AdaBelief Zhuang et al. (2020) replaces Adam's second moment estimates with a squared gradient with the squared difference between the gradient and its running mean to improve convergence and generalization. Lion Chen et al. (2022) uses only sign-based moment updates without per-parameter scaling to reduce memory costs. Adabound Luo et al. (2019) stabilizes learning rates between dynamic lower and upper thresholds to transition from adaptive behavior to SGD-like stability. RAdam Liu et al. (2019a) introduces rectification for the variance of adaptive learning rates, improving training stability in early iterations. Defazio et al. (2024) introduced a schedule-free optimizer that requires no additional hyper-parameters over standard optimizers with momentum. Interestingly, Zhang et al. (2024c) finds that block structures of diagonal Hessians can help reduce memory costs without harming performance. All of these variants feature empirical estimations of first and second moments but with changes like moment center, regularization, etc., which implies that the proposed method can also be translated to the zeroth-order version of these variants. Additionally, Pethick et al. (2025) explores leveraging linear minimization oracles to adapt to loss landscapes without Adam-style updates.

### 2.3 Acceleration by Adam

Compared with first-order optimizers, second-order informed optimizers consider second-order information in the process of gradient calculation. The design of Adam mimics Newton's method with second derivatives. Specifically, the second moment can be viewed as a rough approximation to the inverse Hessian. Lines of work provide analytic or numeric support for Adam's near-diagonal Hessians estimation in deep learning. Das et al. (2024) formalizes that diagonally-dominant Hessians

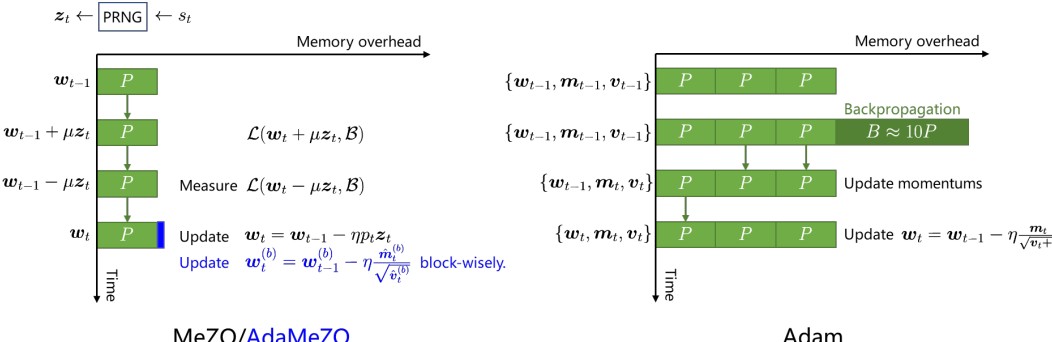

Figure 2: Memory used for storing parameters of AdaMeZO, MeZO and Adam. Arrows indicate in-place parameter modifications.

make Adam mathematically faster. Zhang et al. (2024b) finds block-diagonal Hessians in real neural networks and shows Adam outperforms SGD precisely due to this structure. Empirically, Elsayed et al. (2024) measures strong diagonal dominance in MLP Hessians. Gui et al. (2021) demonstrates that over-parameterization further drives the Hessian toward a diagonal form. Interestingly, Ghorbani et al. (2019) found that the Hessian spectra of deep neural networks become stable after less than $1\%$ training step budget. Kunstner et al. (2023) finds that Adam's great performance could be attributed to its similarity to sign descent with momenta.

## 3 METHODS

In this section, we first introduce the classic approach of the forward pass-only gradient estimator known as SPSA, which is the foundation of MeZO Malladi et al. (2023). Then, we will explain why a direct splicing of SPSA and Adam-style update rule will cause excessive memory usage and how our technique can prevent the issue.

### 3.1 PRELIMINARIES

**Definition 3.1** (Simultaneous Perturbation Stochastic Approximation, SPSA Spall (1992)). *Given a model with weight $w_t$ at step $t$ and objective function $\mathcal{L}$, SPSA estimates the gradient on a batch $\mathcal{B}$ with perturbation scale $\mu > 0$ and random direction $z_t$ as*

$$g_t = p_t z_t, \quad p_t = \frac{\mathcal{L}(w_{t-1} + \mu z_t, \mathcal{B}_t) - \mathcal{L}(w_{t-1} - \mu z_t, \mathcal{B}_t)}{2\mu}. \quad (1)$$

Following prior works, we assume $z \sim \mathcal{N}(\mathbf{0}, I_d)$. It can be shown that $g_t \to \nabla \mathcal{L}(w_t, \mathcal{B}_t)$ as $\mu \to 0$, and is treated as an unbiased gradient estimator with a sufficiently small perturbation scale $\mu$. With an SGD-styled update rule of $w_t \leftarrow w_t - \eta g_t$, modifying the model parameters for gradient estimation and model update can be done in-place as shown in Figure 2. MeZO runs quickly on GPUs since they can spawn random gradients the size of 77.5 B within a second[1]. As a result, MeZO generates little information in memory for fine-tuning compared to backpropagation. The method is shown to yield competitive performance Malladi et al. (2023).

### 3.2 RECOVERING TRUNCATED FIRST MOMENT WITHOUT ADDITIONAL MEMORY

Using the first moment constructed by history gradients with EMA as updates is a widely used technique to cancel out instantaneous gradient noise, hence promoting convergence. Common first-order algorithms require an additional trunk of memory of size $P$ to store the current first moment $m_t$ as follows:

$$m_t \leftarrow (1 - \beta_1)m_t + g_t, \quad w_t \leftarrow w_t - \eta m_t.$$

---

[1]https://developer.nvidia.com/curand

---

**Algorithm 1** $h$-MeZO

---

**Input:** Initialized model parameters $\boldsymbol{w}_0 \in \mathbb{R}^d$, loss function $\mathcal{L} : \mathbb{R}^d \to \mathbb{R}$, step budget $T$, perturbation scale $\mu$, learning rate $\eta$, horizon $h$, first EMA ratio $\beta_1$
**Output:** Trained model parameters $\boldsymbol{w}_T$
`seeds, projs ← [], []`
**for** $t = 1, \ldots, T$ **do**
    Sample batch $\mathcal{B}_t$ and random seed $s$
    Reset the PRNG with random seed $s$, spawn $\boldsymbol{z}_t \sim \mathcal{N}(\boldsymbol{0}, I_d)$
    Estimate $p_t$ using Equation (1)                 *# in-place model perturbation*
    `seeds.append(`$s$`), projs.append(`$p_t$`)`
    $\boldsymbol{w}_t \leftarrow \boldsymbol{w}_t$
    **for** $\tau = 1, \ldots, h$ **do**
        $p \leftarrow$ `projs`$[-\tau]$, $s \leftarrow$ `seeds`$[-\tau]$
        Reset the PRNG with random seed $s$, spawn $\boldsymbol{z} \sim \mathcal{N}(\boldsymbol{0}, I_d)$
        $\boldsymbol{w}_t \leftarrow \boldsymbol{w}_t - \eta \beta_1^{\tau-1} p \boldsymbol{z}$
    **end for**
**end for**

---

However, the MeZO-styled in-place parameter update allows the first moment to be approximated without storing history gradients. Specifically, we unroll the recursion, set a hyperparameter, the horizon $h$, and discard the outdated gradients computed more than $h$ steps ago, then employ the similar in-place parameter update process as in MeZO as Equation (2) and detailed in Algorithm 1.

$$\boldsymbol{m}_t = \boldsymbol{g}_t + \beta_1 \boldsymbol{g}_t + \beta_1^2 \boldsymbol{g}_{t-2} + \cdots + \beta_1^{t-h-1} \boldsymbol{g}_{t-h-1}. \tag{2}$$

*Remark* 3.2. The idea behind Algorithm 1 is that the share of a history gradient $\boldsymbol{g}_{t-t'}$ decays quickly. As an example, after sufficiently long steps, the share of $\boldsymbol{g}_{t-10}$ approximates $0.9^{10}/(1/(1-0.9)) \approx 0.0387$ at $\beta_1 = 0.9$. It implies that the key components of the first moment are several of the most recent gradients, while the rest are relatively safe to be omitted. Supported by the PRNG as a coder of the random gradients, Algorithm 1 can use truncated first moments without additional memory.

### 3.3 SECOND MOMENT INFORMED UPDATES WITHOUT ADDITIONAL MEMORY

We can similarly recover a truncated second moment. However, bringing them into the update will still be costly. We investigate the issue and present our solution in this subsection.

An Adam-style update rule can be expressed as follows:

$$\boldsymbol{m}_t \leftarrow (1-\beta_1)\boldsymbol{m}_t + \boldsymbol{g}_t, \quad \boldsymbol{v}_t \leftarrow (1-\beta_2)\boldsymbol{v}_t + \boldsymbol{g}_t \odot \boldsymbol{g}_t, \quad \boldsymbol{w}_t \leftarrow \boldsymbol{w}_t - \eta \frac{\boldsymbol{m}_t}{\sqrt{\boldsymbol{v}_t + \epsilon}}. \tag{3}$$

We can notice that the update needs the complete $\boldsymbol{v}_t$ beforehand. Specifically, for an update, we will need to construct $\boldsymbol{m}_t$ after computing $\boldsymbol{v}_t$ as

$$\boldsymbol{v}_t = \boldsymbol{g}_t \odot \boldsymbol{g}_t + \beta_2(\boldsymbol{g}_t \odot \boldsymbol{g}_t) + \beta_2^2(\boldsymbol{g}_{t-2} \odot \boldsymbol{g}_{t-2}) + \cdots + \beta_2^{t-h}(\boldsymbol{g}_{t-h} \odot \boldsymbol{g}_{t-h}),$$

as shown in Figure 3. As a result, we will need a memory trunk of size $P$ to store $\boldsymbol{v}_t$ before unrolling $\boldsymbol{m}_t$, doubling the memory requirement. This method is employed in Zhao et al. (2024a). To address this issue, we propose to cache the random states rather than the seeds during the training.

#### 3.3.1 FINE-SCALED RANDOM STREAM GENERATION BY STATE CACHING

A formal expression of how concurrent PRNG algorithms generate random number streams is as Algorithm 2, with algorithm-specified and deterministic state update function $F$ and extractor $O$.

For prior practices including Malladi et al. (2023); Zhao et al. (2024a;b), the above process guarantees identical and complete $\boldsymbol{z}_t$ for gradient estimation and gradient updating is obtained by caching the seed $s$, so that in-place gradient estimation and parameter updating without additional memory access functions. However, caching the random state $S$ offers the possibility to *jump* to a specified position in a random number stream. It allows the PRNG to faithfully continue a particular random

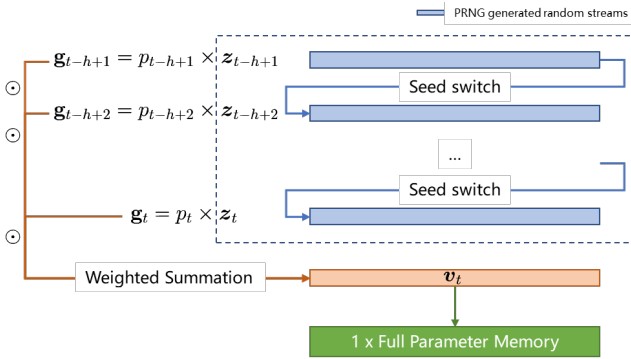

Figure 3: Caching the complete second moment recover before the first moment unrolling, doubling the memory requirement.

---

**Algorithm 2** PRNG

---

**Input:** Seed $s$
**Output:** Random number streams $r_n$
Initial state mapper $I$ maps the random seed $s$ to the initial state $S_0$.
**while** No stop signal **do**
    PRNG outputs random stream $\{r_n\}$ by the recurrence

$$r_n = O(S_{n-1}), \quad S_n = F(S_{n-1}), \tag{4}$$

**end while**

---

stream from wherever it is left over, making it more flexible than seed-caching. Code examples for this feature can be found in Section C.

An illustration of Block-wise moment approximation is as Figure 4. A parameter block partition $\boldsymbol{w} = \{\boldsymbol{w}^{(1)}, \boldsymbol{w}^{(2)}, \ldots, \boldsymbol{w}^{(b)}\}$ is prepared at the beginning of a parameter update. We first start at the first block. Processing of the first block is the same as Figure 3, which is how prior works exploit PRNGs. However, after the first random direction block $\boldsymbol{z}_{t-t'}^{(1)}$ is spawned, AdaMeZO records $S_n$, the corresponding random state, for each seed. When the spawning of the first block from each of the seeds within the horizon is finished, AdaMeZO skips the initial state mapping in Algorithm 2, and loads the cached $S_n$ to the PRNG for the next block, so that the contiguous random stream is generated rather than starting over again from the first output of the random stream. The process loops until all the blocks finish their update.

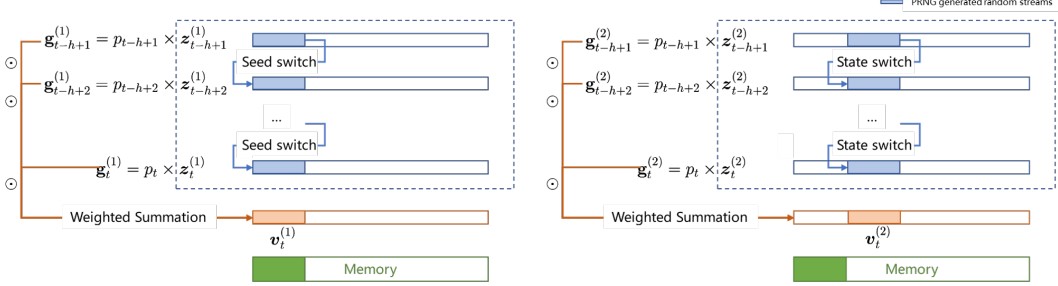

Figure 4: Block-wise moment approximation in AdaMeZO. $\odot$ denotes the Hadamard product.

### 3.3.2 ADAM-STYLE UPDATES WITH ZEROTH-ORDER GRADIENTS

With random state caching, we can update models according to Equation (3) block-wise, which is impossible for seed caching as employed by previous works, since seed caching only allows the random stream to be spawned from the initial digit. However, we need some warm-up steps to

accumulate history gradients before estimating finite-horizon moments. Due to page limits, we elaborate the process as Algorithm 3 in the Appendix.

*Remark* 3.3. It is worth mentioning that caching random states adds a bit-level memory cost compared to caching seeds. In Philox Salmon et al. (2011), the default choice of CUDA PRNG, random state $S$ consists of a 64-bit random seed, a 64-bit subsequence identifier, and a 64-bit offset. Mersenne Twister Matsumoto & Nishimura (1998), the default choice of CPU PRNG, maintains similar information as random states. Therefore, caching the random states incurs a negligible additional memory cost at the bit level compared to caching seeds.

*Remark* 3.4. Though the first and second moments do not go into the memory, recovering them requires a temporary additional memory trunk, whose size scales to the size of the block, corresponding to the $\delta P$ term in Table 1. A natural block strategy is the different layers of the model. If the model consists of 32 layers, the additional memory introduced in the second moment approximation is roughly $2/32P$ ($1/32P$ for $\boldsymbol{m}_t^{(b)}$ and $\boldsymbol{v}_t^{(b)}$ each). However, the block can be made as small as consisting of only 1 parameters. Therefore, AdaMeZO can theoretically approximate the moments by performing frequent random state dumps and loads without incurring additional memory requirements.

## 4 THEORY

We employ the following widely adopted assumptions to facilitate an analysis.

**Assumption 4.1** (L-smooth). *For any weight vector $\boldsymbol{w}_1, \boldsymbol{w}_2 \in \mathbb{R}^d$, for a $0 < L < \infty$ it holds that*

$$\mathcal{L}(\boldsymbol{w}_2) \leq \mathcal{L}(\boldsymbol{w}_1) + \langle \nabla \mathcal{L}(\boldsymbol{w}_1), \boldsymbol{w}_2 - \boldsymbol{w}_1 \rangle + \frac{L}{2} \|\boldsymbol{w}_2 - \boldsymbol{w}_1\|_2^2.$$

**Assumption 4.2** (Bounded gradient variance). *The stochastic gradient $\nabla \mathcal{L}(\boldsymbol{w}_t, \mathcal{B}_t)$ has no bias and $\sigma^2$ variance due to batch stochasticity, specifically*

$$\mathbb{E}_{\mathcal{B}_t} [\nabla \mathcal{L}(\boldsymbol{w}_t, \mathcal{B}_t)] - \nabla \mathcal{L}(\boldsymbol{w}_t) = 0, \tag{5}$$

$$\mathbb{E}_{\mathcal{B}_t} [\|\nabla \mathcal{L}(\boldsymbol{w}_t, \mathcal{B}_t)\|_2^2] - \|\nabla \mathcal{L}(\boldsymbol{w}_t)\|_2^2 \leq \sigma_t^2, \quad \sigma_t < \infty.$$

**Assumption 4.3** (Bounded second moment, Zhao et al. (2024b)). *Each entry of $\Sigma_t$ lies in the range $[s_l, s_u]$ with $0 < s_l < s_u < \infty$.*

**Assumption 4.4** (Weak gradient stationary within horizon). *The gradients within the moment horizon $h$ are stationary, specifically,*

$$\|\mathbb{E}_{0 \leq i \leq h}[\nabla \mathcal{L}(\boldsymbol{w}_{t-i})] - \nabla \mathcal{L}(\boldsymbol{w}_t)\|_2^2 \leq \mathcal{O}(\beta_1 L \eta).$$

**Lemma 4.5** (Magnus et al. (1978)). *Let $A$ and $B$ be two symmetric matrices, $\boldsymbol{z} \sim \mathcal{N}(\boldsymbol{0}, \boldsymbol{I}_d)$. Define $\boldsymbol{x} = \boldsymbol{z}^\top A \boldsymbol{z} \boldsymbol{z}^\top B \boldsymbol{z}$, then it holds that*

$$\mathbb{E}_{\boldsymbol{z}}[\boldsymbol{x}] = (\mathrm{tr}A)(\mathrm{tr}B) + 2\mathrm{tr}(AB).$$

**Assumption 4.6** (Local $r$-effective rank, Malladi et al. (2023)). *Let $G(\boldsymbol{w}_t) := \max_{\mathcal{B}, |\mathcal{B}|=1} \|\nabla \mathcal{L}(\boldsymbol{w}_t, \mathcal{B})\|$. There is a matrix $\mathcal{H}(\boldsymbol{w}_t) \preceq L I_d$ satisfying:*

  1. *For all $\boldsymbol{w}$ such that $\|\boldsymbol{w} - \boldsymbol{w}_t\|_2 \leq \eta d G(\boldsymbol{w}_t)$, it holds that $\nabla^2 \mathcal{L}(\boldsymbol{w}) \preceq H(\boldsymbol{w}_t)$.*

  2. *The effective rank of $H(\boldsymbol{w}_t)$, specifically, $\mathrm{tr}(\mathcal{H}(\boldsymbol{w}_t))/\|\mathcal{H}(\boldsymbol{w}_t)\|_{op}$, is at most $r$.*

We present a convergence bound on the assumption of non-convex optimization, details in Section E. The bound resembles the structure from Zhao et al. (2024b).

**Theorem 4.7.** *With a sufficiently small learning rate $\eta$, AdaMeZO converges to a stationary point with*

$$\mathbb{E}\left[\frac{1}{T}\sum_{t=1}^{T} \|\nabla \mathcal{L}(\boldsymbol{w}_t)\|_2^2\right] \leq \frac{L\sigma^2}{\sqrt{T}} + \frac{2}{T\sqrt{T}} + \frac{T_w \sigma^2}{T\sqrt{T}} + \mathcal{O}(\beta_1 L \eta),$$

*where $\sigma$ captures the batch gradient variance in first-order senses, and $T_w$ denotes the number of warmup steps detailed in Algorithm 3.*

Detailed proof can be found in Section E. The above result shows that after $T = \mathcal{O}(\epsilon^2)$ steps, AdaMeZO converges to a small neighborhood of a stationary point satisfying $\mathbb{E}\left[\|\frac{1}{T}\sum_{t=1}^{T} \nabla \mathcal{L}(\boldsymbol{w}_t)\|_2^2\right] < \epsilon$.

Table 2: Results on RoBERTa-large over language tasks with $k = 16$.

| Task | SST-2 | SST-5 | SNLI | MNLI | RTE | TREC | Average |
|------|-------|-------|------|------|-----|------|---------|
| Type | —- sentiment —- | | - natural language inference - | | | – topic – | |
| Zero-shot | 79.0 | 35.5 | 50.2 | 48.8 | 51.4 | 32.0 | 49.4 |
| FO ($12 \times$ memory) | 91.8 | 47.5 | 77.5 | 70.0 | 66.4 | 85.0 | 73.0 |
| MeZO | 90.6 | 44.1 | **67.3** | 58.1 | 61.6 | 67.3 | 64.8 |
| | (1.4) | (1.0) | (3.1) | (1.1) | (1.3) | (2.7) | – |
| MeZO-switch | 90.6 | 44.3 | **67.3** | 58.0 | 61.6 | 67.0 | 64.8 |
| | (1.6) | (1.6) | (2.8) | (1.3) | (2.0) | (4.7) | – |
| ***AdaMeZO*** | **90.9** | **45.2** | 66.8 | **58.6** | **63.1** | **71.5** | **66.0** |
| | (0.9) | (2.0) | (2.9) | (1.4) | (2.3) | (5.2) | – |

## 5 EXPERIMENT RESULTS

We present empirical results for AdaMeZO with its baselines in this section. Generally, there are two types of LLMs: 1) encoder-decoder, or masked language models (MLM), such as BERT Devlin et al. (2019) and its variants, and 2) decoder-only, or autoregressive models (ARM), such as GPT, OPT, and LLaMA families. To comprehensively demonstrate the performance of AdaMeZO, we first illustrate the optimization trajectories on toy functions. Then, we test AdaMeZO with baseline algorithms on well-recognized LLMs, including an MLM RoBERTa Liu et al. (2019b), and two ARMs, OPT Zhang et al. (2022a) and Touvron et al. (2023).

### 5.1 TOY FUNCTIONS

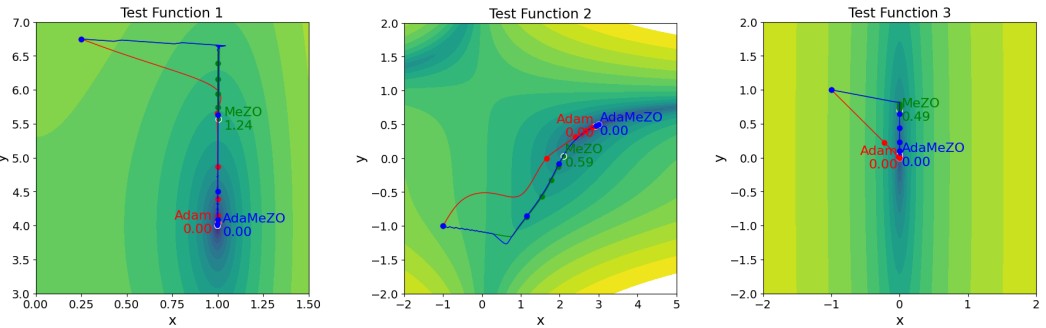

Figure 5: Optimization trajectories on test functions. The loss values at termination are labeled.

It is impractical to visualize trajectories on models with a number of dimensions in billions. However, we can illustrate the optimization trajectories on three 2-dimensional toy functions as in Figure 5 to show how AdaMeZO adapts to heterogeneous curvatures. We test the Adam optimizer Kingma & Ba (2014) implemented in PyTorch Paszke (2019), the vanilla MeZO Malladi et al. (2023), and the proposed AdaMeZO. More details in Section B.4.

In general, we observe that AdaMeZO shares the curvature adaptability of Adam. Although AdaMeZO walks longer paths due to the stochasticity of the gradient direction and warm-up steps, it moves swiftly in areas with small curvatures thanks to the preconditioning of the diagonal Hessian estimator, and the final loss values are comparable to Adam's. In contrast, MeZO struggles with oscillations in low curvature areas and results in worse convergence.

### 5.2 MASKED LANGUAGE MODELS

We compare the performance of AdaMeZO with vanilla MeZO and MeZO-switch—a variant of MeZO where the learning rate is manually adjusted to ensure that its optimization trajectory is longer than that of AdaMeZO. This demonstrates that AdaMeZO's outperformance is not due to MeZO's underfitting, but rather to its adaptability to the loss landscape.

Consistent with previous research Malladi et al. (2023), we conduct experiments on RoBERTa-large 350M on three types of NLP tasks: sentiment, natural language inference, and topic. We sample $k$ examples per class for $k = 16$ (results in Table 2) to demonstrate the training performance under few-shot and many-shot scenarios, respectively. We repeat 4 times on different seeds[2] and report the mean and standard deviation of the corresponding metric for each task. It is found that:

**AdaMeZO yields better performance.** Averaged across all tasks, AdaMeZO achieves a **1.2%** absolute accuracy improvement to MeZO on average, with particularly strong gains in tasks like RTE (**1.5%**), TREC (**4.2%**).

## 5.3 AUTOREGRESSIVE MODELS

Table 3: Main results on OPT-1.3B over language tasks. Avg (w.o S,D) indicates the average performance value except SQuAD and DROP.

| Task Type | SST-2 | RTE | CB | BoolQ | WSC | WIC | MultiRC | COPA | ReCoRD | SQuAD | DROP | Avg | Avg (w.o S,D) |
|---|---|---|---|---|---|---|---|---|---|---|---|---|---|
| | ——————— classification ——————— | | | | | | | – multiple choice – | | — generation — | | | |
| Zero-shot | 53.5 | 53.4 | 39.2 | 45.5 | 43.2 | 57.5 | 45.4 | 75.0 | 70.5 | 27.2 | 11.1 | 47.4 | 53.6 |
| FO (12 × memory) | 90.9 | 64.0 | 77.2 | 64.4 | 52.8 | 62.3 | 65.2 | 74.0 | 69.1 | 80.4 | 28.2 | 66.2 | 68.9 |
| | (1.2) | (10.7) | (7.9) | (9.3) | (2.0) | (1.9) | (6.0) | (2.9) | (1.2) | (1.5) | (1.7) | – | – |
| MeZO | 90.9 | 52.5 | 65.5 | 61.8 | 51.1 | **58.6** | 53.7 | 74.5 | 70.6 | 73.3 | 22.8 | 61.4 | 64.4 |
| | (0.3) | (1.5) | (6.9) | (2.1) | (8.4) | (1.4) | (2.2) | (3.6) | (1.0) | (0.2) | (0.6) | – | – |
| MeZO-switch | 91.0 | 53.8 | 68.7 | 61.9 | 52.1 | 58.3 | 54.9 | **75.5** | 71.0 | 73.7 | 24.3 | 62.3 | 65.2 |
| | (0.6) | (1.6) | (2.3) | (0.6) | (7.6) | (1.6) | (1.5) | (3.6) | (1.2) | (1.2) | (1.3) | – | – |
| HiZOO | 90.9 | **54.5** | 63.3 | 62.7 | 49.4 | 58.4 | 55.4 | 74.0 | 70.8 | 74.5 | 24.5 | 61.7 | 64.4 |
| | (1.0) | (1.6) | (8.5) | (1.6) | (6.9) | (0.4) | (1.7) | (1.8) | (0.8) | (0.4) | (0.5) | – | – |
| *AdaMeZO* | **91.6** | 54.3 | **69.6** | **63.2** | **53.5** | 58.4 | 55.9 | **75.5** | 71.1 | **76.1** | 24.6 | 63.1 | **65.9** |
| | (0.3) | (3.1) | (1.4) | (1.6) | (7.8) | (1.6) | (0.7) | (4.0) | (1.3) | (0.7) | (1.0) | – | – |

Table 4: Main results on LLaMA-3B over language tasks.

| Task Type | SST-2 | RTE | CB | BoolQ | WSC | WIC | MultiRC | COPA | ReCoRD | SQuAD | DROP | Avg | Avg (w.o S,D) |
|---|---|---|---|---|---|---|---|---|---|---|---|---|---|
| | ——————— classification ——————— | | | | | | | – multiple choice – | | — generation — | | | |
| Zero-shot | 56.0 | 52.7 | 51.6 | 60.9 | 36.5 | 54.3 | 44.8 | 75.0 | 68.2 | 47.3 | 20.8 | 51.6 | 55.5 |
| FO (12 × memory) | 92.5 | 73.9 | 85.6 | 65.9 | 57.8 | 67.1 | 70.6 | 75.7 | 68.6 | 83.9 | 32.2 | 70.3 | 73.1 |
| | (0.7) | (5.4) | (6.9) | (7.3) | (7.6) | (0.7) | (1.8) | (2.6) | (1.0) | (0.3) | (1.8) | – | – |
| MeZO | 84.5 | 53.2 | 64.7 | 62.6 | 50.4 | 54.6 | 52.6 | 77.2 | 70.0 | 79.2 | 26.8 | 61.4 | 63.3 |
| | (4.9) | (0.7) | (2.6) | (0.7) | (11.3) | (0.3) | (2.5) | (2.0) | (0.9) | (0.9) | (0.5) | – | – |
| MeZO-switch | 86.6 | 54.1 | 65.5 | 63.2 | 51.6 | 54.7 | 54.7 | 78.7 | 70.4 | **80.4** | 27.6 | 62.5 | 64.4 |
| | (4.5) | (1.5) | (0.9) | (0.3) | (12.2) | (1.0) | (0.6) | (2.2) | (0.6) | (0.9) | (0.6) | – | – |
| HiZOO | 92.2 | 54.1 | 65.1 | 63.7 | 52.8 | **54.9** | 56.5 | **82.5** | **71.5** | 18.5 | 6.1 | 56.1 | 65.9 |
| | (0.5) | (0.3) | (2.2) | (0.3) | (5.4) | (1.7) | (0.7) | (0.5) | (0.6) | (1.9) | (1.2) | – | – |
| *AdaMeZO* | **92.6** | 54.4 | 66.0 | 64.6 | 54.5 | 54.9 | 56.9 | 81.2 | 71.3 | 80.4 | 28.1 | 64.1 | **66.3** |
| | (0.5) | (1.5) | (1.4) | (2.6) | (7.5) | (1.6) | (1.0) | (3.2) | (0.9) | (1.8) | (1.1) | – | – |

Then we extend our investigation to two autoregressive models: OPT-1.3B (results in Table 3) and LLaMA-3B (results in Table 4). Experimental results demonstrate AdaMeZO's consistent superiority across diverse language tasks for both OPT-1.3B and LLaMA-3B models. It is found that:

**AdaMeZO's superior performance scales up to billion-level LLMs.** On OPT-1.3B, AdaMeZO surpasses MeZO and MeZO-switch in all but one instance. AdaMeZO achieves a **1.7%** absolute accuracy improvement to MeZO on average, with particularly strong gains in tasks like CB (**4.1%**), SQuAD (**2.8%**), WSC (**2.4%**), and MultiRC (**2.2%**). For LLaMA-3B, AdaMeZO further extends its lead, achieves a **2.7%** absolute accuracy improvement to MeZO on average, with particularly strong gains in tasks like SST2 (**8.1%**), MultiRC (**4.3%**), WSC (**4.1%**), COPA (**4.0%**). AdaMeZO's advantage remains in larger models, such as OPT-13B, as reported in Table 5.

---

[2]Two kinds of seeds are used in our work. The first kind is fed into PRNGs to generate random gradient directions. The second kind is the seeds for the random seed sampler to sample the seeds of the first kind. The second kind of seed can be considered to play the same role as "random seeds" in general works. We refer to the second kind here.

Table 5: Main results on OPT-13B over language tasks. OOM indicates that HiZOO encountered an out-of-memory error. A cell is marked as OOM if any of the evaluation seeds trigger an OOM. The official HiZOO implementation only supports single-GPU training, and the required memory footprint exceeds the capacity of our largest GPU (A100 80GB), leading to OOM failures.

| Task Type | SST-2 | RTE | CB | BoolQ | WSC | WIC | MultiRC | COPA | ReCoRD | SQuAD | DROP | Avg | Avg (w.o S,D) |
|---|---|---|---|---|---|---|---|---|---|---|---|---|---|
| | | | classification | | | | | – multiple choice – | | — generation — | | | |
| Zero-shot | 58.8 | 59.6 | 46.4 | 59.0 | 38.5 | 55.0 | 46.9 | 80.0 | 81.2 | 46.2 | 14.6 | 53.2 | 58.3 |
| FO ($12 \times$ memory) | 92.0 | 70.8 | 83.9 | 77.1 | 63.5 | 55.0 | 71.1 | 79.0 | 74.1 | 84.9 | 31.3 | 71.1 | 74.0 |
| MeZO | 92.1 | 60.4 | 67.8 | 65.5 | 56.6 | 54.9 | 56.7 | **87.0** | 80.2 | 82.1 | 30.6 | 66.7 | 69.0 |
| | (0.5) | (0.6) | (1.4) | (3.0) | (7.9) | (1.7) | (0.8) | (1.1) | (1.0) | (1.3) | (1.5) | – | – |
| MeZO-switch | 92.6 | 61.6 | 66.9 | 66.2 | 56.9 | 55.4 | 57.5 | 86.0 | **80.5** | 83.4 | 30.5 | 67.0 | 69.3 |
| | (0.2) | (2.5) | (1.0) | (3.7) | (8.4) | (0.7) | (0.5) | (2.7) | (1.1) | (0.8) | (0.9) | – | – |
| HiZOO | 91.5 | 62.5 | **68.2** | OOM | 56.4 | 55.4 | 57.5 | 86.2 | 80.0 | 83.5 | OOM | – | – |
| | (1.3) | (3.7) | (2.2) | – | (8.3) | (1.4) | (0.6) | (0.9) | (1.4) | (1.2) | – | – | – |
| *AdaMeZO* | **92.7** | **63.0** | 67.8 | **70.6** | **58.4** | **55.8** | **58.3** | **87.0** | 80.1 | 83.7 | **31.0** | **68.0** | **70.4** |
| | (0.5) | (6.1) | (2.5) | (3.6) | (7.5) | (0.6) | (0.3) | (1.1) | (0.7) | (1.3) | (0.9) | – | – |

Table 6: Performance comparison with different $(\beta_1, \beta_2)$.

| $(\beta_1, \beta_2)$ | SST2 | COPA | SQuAD |
|---|---|---|---|
| (0.7, 0.9) | 91.6 (0.3) | 75.5 (4.0) | 76.1 (0.7) |
| (0.7, 0.99) | 90.9 (0.9) | 75.3 (2.9) | 75.6 (0.9) |
| (0.6, 0.9) | 91.1 (0.6) | 75.8 (2.9) | 75.6 (1.2) |
| (0.8, 0.9) | 91.5 (0.7) | 74.3 (2.9) | 75.6 (1.5) |
| (0.7, 0.0), mSGD | 90.9 (0.9) | 75.3 (2.9) | 75.6 (0.9) |
| (0.0, 0.0), MeZO | 90.9 (0.3) | 74.5 (3.6) | 73.3 (0.2) |

## 5.4 HYPERPARAMETER ANALYSIS

Similar to Adam, two new hyperparameters $(\beta_1, \beta_2)$ are introduced into the algorithm. We report AdaMeZO's performance with different hyperparameter settings as in Table 6. It can be observed that the first moments can improve the performance of MeZO, and the second moments can further improve it. The performance gain is robust against reasonable choices of the hyperparameter $(\beta_1, \beta_2)$.

## 6 CONCLUSION, LIMITATIONS AND FUTURE WORKS

In this work, we introduce AdaMeZO, the first ZOO that incorporates Adam-style first and second moments without doubling or tripling the memory requirements of the original MeZO. This is achieved by estimating truncated moments and performing more refined operations on PRNGs. We provide theoretical analysis and empirical evaluations. Visualizations show that AdaMeZO adapts to complicated loss landscapes without excessively consuming additional memory. Experiments on well-recognized models show that AdaMeZO reaches on-par performance using fewer forward passes and can continue to lower loss values before reaching identical terminal conditions. The paper's limitations are as follows.

We have captured the gradient drift across different steps using a big $\mathcal{O}$ constant related to L-smoothness and learning rate. Although finite moment horizons may help to keep the estimations less biased, we did not attempt to explicitly capture the gap, which is a future research direction.

AdaMeZO estimates second moments at a small cost, but they are inaccurate. The reason is two-fold: 1) AdaMeZO runs on zeroth-order gradient estimations, and 2) a smaller $\beta_2$ to guarantee that the discarded part contributes only a small share. Future investigations into more accurate second-moment estimations could improve performance.

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

## A    ADDITIONAL RELATED WORKS

In addition to MeZO, numerous subsequent excellent works have emerged to enhance the vanilla version. Jiang et al. (2024) incorporates uncertain moments estimations to promote convergence. Zhao et al. (2024a) invokes Adam-style update rules for better performance. Zhao et al. (2024b) estimates the diagonal Hessian with a three-point second derivative estimation admitted by a third forward pass for each step. Liu et al. (2024); Guo et al. (2024) proposed to insert sparsity for better performance. Chen et al. (2024); Sun et al. (2025) exploits the low-rank property for better performance. Chen et al. (2025a) proposes a hybrid optimizer for efficiency trade-offs. Tan et al. (2025) explores a layer-wise adaptation to speed up zeroth-order fine-tuning. Chen et al. (2025b) investigates memory-efficient zeroth-order fine-tuning from a perspective of subspace optimization. Yu et al. introduces a block version of HiZOO, attempting to preserve preconditioning-improved convergence while reducing additional memory access.

## B    DETAILED EXPERIMENT SETTINGS

Other preconditioned MeZO like Zhao et al. (2024a;b) are excluded since they pose substantially additional memory requirements.

### B.1    COMPUTATION RESOURCES

We summarize the computational devices for empirical evaluations in Table 7. We use device 1 for MLM experiments and device 2 for ARM experiments.

Table 7: Summary of computational devices for empirical evaluations.

| Device | OS/CPU/GPU | Python | PyTorch | CUDA | cuDNN |
|---|---|---|---|---|---|
| 1 | Linux 5.10.0, amd64
Intel(R) Xeon(R) Gold 6133 CPU @ 2.50GHz
6x NVIDIA GeForce RTX 3090 GPU | 3.10.13 | 2.3.0 | 12.1 | 8.9 |
| 2 | Linux 4.18.0, x86_64
AMD EPYC 7742 64-Core Processor
4x NVIDIA A100-SXM4-80GB | 3.11.9 | 2.2.0 | 12.1 | 8.9 |

## B.2 FORMAL PSEUDO-CODES FOR ADAMEZO

A formal description of AdaMeZO in pseudo-code is as Algorithm 3.

---

**Algorithm 3** AdaMeZO

---

**Input:** Initialized model parameters $w_0 \in \mathbb{R}^d$, loss function $\mathcal{L} : \mathbb{R}^d \to \mathbb{R}$, step budget $T$, perturbation scale $\mu$, learning rate $\eta$, horizon $h$, first EMA ratio $\beta_1$, second EMA ratio $\beta_2$, block strategy $B(w) = \{w^{(1)}, \ldots, w^{(b)}\}$, cancel factor $\beta_v$, warm-up steps $T_w$

**Output:** Trained model parameters $w_T$

```
seeds, projs ← [], []
```

**for** $t = 1, \ldots, T$ **do**

    Sample batch $\mathcal{B}_t$ and random seed $s$

    Reset the PRNG with random seed $s$, spawn $z_t \sim \mathcal{N}(\mathbf{0}, I_d)$

    Estimate $p_t$ using Equation (1)         # *in-place model perturbation*

    ```seeds.append(s), projs.append(p_t)```

    $w_t \leftarrow w_t$

    **if** $t > T_w$ **then**

        ```states ← [None] * (h, b)```

        **for** $\tau_b = 1, \ldots, b$ **do**

            $m, v \leftarrow \mathbf{0}, \mathbf{0}$

            **for** $\tau_h = 1, \ldots, h$ **do**

                $p \leftarrow$ ```projs[-τ_h]```

                **if** ```states[τ_h, τ_b] == None``` **then**

                    $s \leftarrow$ ```seeds[-τ_h]```

                    Reset the PRNG with random seed $s$, spawn $z \sim \mathcal{N}(\mathbf{0}, I_{|w^{(\tau_b)}|})$

                **else**

                    Load ```states[τ_h, τ_b]``` to PRNG, spawn $z \sim \mathcal{N}(\mathbf{0}, I_{|w^{(\tau_b)}|})$

                **end if**

                Save PRNG state to ```states[τ_h, τ_b]```

                $m \leftarrow m + \beta_1^{\tau_h - 1} p z$

                $v \leftarrow v + \beta_2^{\tau_h - 1} p^2 (z \odot z)$

            **end for**

        **end for**

        $w_t^{(\tau_b)} \leftarrow w_t^{(\tau_b)} - \eta \beta_v \frac{m}{\sqrt{v + \epsilon}}$

    **else**

        Reset the PRNG with random seed $s$, spawn $z \sim \mathcal{N}(\mathbf{0}, I_d)$

        $w_t \leftarrow w_t - \eta p_t z$

    **end if**

**end for**

---

## B.3 DETAILED SETTINGS FOR FIGURE 1

For (a), the learning rate is 1e-6, with 16 training samples per class. For (b) and (c), the learning rate is 1e-7, with 1000 training samples in total. We set $\beta_1 = 0.7, \beta_2 = 0.9, h = 10$, so that AdaMeZO discards only a small part by truncating the moments and admits a smoother second moment estimation compared to the first. With an abuse of context, the choice of $(\beta_1, \beta_2)$ falls into the suggested area $0 < \beta_1 < \sqrt{\beta_2} < 1$ by Zhang et al. (2022b). Fine-tuning terminates when either of the following happens.

1. Measure evaluation loss per 100 steps. Evaluation loss does not drop for 5 continual measures.

2. Number of steps exceeds 40000.

## B.4 DETAILED SETTINGS FOR SECTION 5.1

The expressions of the test functions are

1. $f_1(x, y) = 8(x - 1)^2(1.3x^2 + 2x + 1) + 0.5(y - 4)^2$ Zhao et al. (2024b).

2. $f_2(x, y) = (1.5 - x + xy)^2 + (2.25 - x + xy^2)^2 + (2.625 - x + xy^3)^2$ Beale (1958).

3. $f_3(x, y) = 100x^2 + y^2$.

Specifications on implementations are as Table 8. The setting follows on the following rule:

1. Set the learning rate of Adam to $0.01$,

2. Tune the learning rate for ZO optimizers so that the trajectory lengths are comparable to Adam's. We allow a longer trajectory ($< 1.6\times$) for ZO optimizers.

For MeZO and AdaMeZO, we allow only 2 seeds coding 2 gradient directions. This is to capture the situation where the number of steps, equivalently the total number of explored gradient directions (in thousands), is usually less than the number of dimensions of the LLMs (in billions).

Table 8: Specifications for toy functions.

|       | Adam | | MeZO | | AdaMeZO | | # steps | Initialization |
|-------|------|--------|-------|--------|---------|--------|---------|----------------|
|       | lr   | length | lr    | length | lr      | length |         |                |
| $f_1$ | 0.01 | 3.0227 | 0.01  | 4.6659 | 0.01    | 4.5078 | 600     | $(0.2, 6.75)$  |
| $f_2$ | 0.01 | 4.3597 | 0.002 | 5.5405 | 0.002   | 5.3207 | 2500    | $(-1, -1)$     |
| $f_3$ | 0.01 | 1.4142 | 0.01  | 1.4243 | 0.01    | 1.8577 | 500     | $(-1, 1)$      |

Trajectories in higher resolutions and 3D views of the loss landscapes are as Figure 6.

### B.5   DETAILED SETTINGS FOR SECTION 5.2 AND SECTION 5.3

Table 9: Hyperparameter settings.

|         | $B$ | $T$             | $\eta$           | $q$ | $\mu$            | $(\beta_1, \beta_2)$ |
|---------|-----|-----------------|------------------|-----|------------------|----------------------|
| Table 2 | 16  | $1 \times 10^5$ | $1 \times 10^{-6}$ | 5 | $1 \times 10^{-3}$ | $(0.7, 0.9)$         |
| Table 3 | 16  | $4 \times 10^4$ | $1 \times 10^{-7}$ | 5 | $1 \times 10^{-3}$ | $(0.7, 0.9)$         |
| Table 4 | 16  | $4 \times 10^4$ | $1 \times 10^{-7}$ | 5 | $1 \times 10^{-3}$ | $(0.7, 0.9)$         |

Fine-tuning terminates when either of the following conditions is met.

1. Measure evaluation loss per 100 steps. Evaluation loss does not drop for $q$ continual measures.

2. Number of steps exceeds $T$.

## C   CODE SNIPPETS FOR BLOCK-WISE GRADIENT GENERATION

Previous works call PRNG by the following codes.

```
torch.manual_seed(seed)
z = torch.normal(
    mean=0,
    std=1,
    size=param.data.size(),
    dtype=param.data.dtype,
    device=param.device,
)
```

In this work, for block-wise gradient generation, we wish the PRNG to skip the random stream belonging to prior blocks. Therefore, we directly feed random states into the PRNG, thereby skipping the initialization step implied by `manual_seed(seed)`. A snippet to realize the feature is as follows.

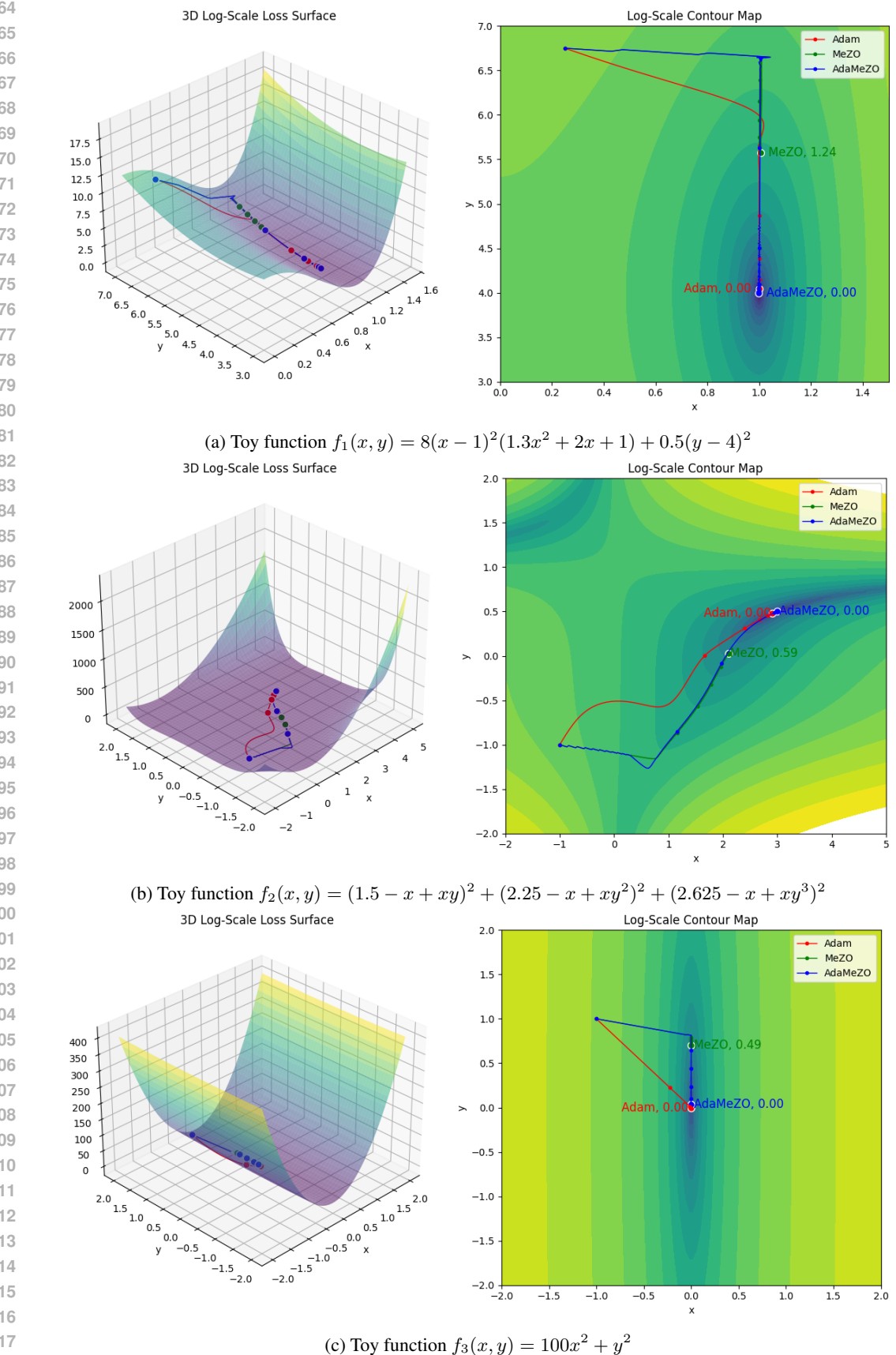

(a) Toy function $f_1(x, y) = 8(x-1)^2(1.3x^2 + 2x + 1) + 0.5(y-4)^2$

(b) Toy function $f_2(x, y) = (1.5 - x + xy)^2 + (2.25 - x + xy^2)^2 + (2.625 - x + xy^3)^2$

(c) Toy function $f_3(x, y) = 100x^2 + y^2$

Figure 6: Loss landscapes of the toy functions and optimization trajectories.

```
1  self.g = torch.Generator(device='cuda')
2  self.g.set_state(state)        # scheduled state
3  z = torch.normal(
4      mean=0,
5      std=1,
6      size=param.data.size(),
7      dtype=param.data.dtype,
8      device=param.device,
9      generator=self.g,
10 )
11 state = self.g.get_state(state)
```

# D  ADDITIONAL EXPERIMENT RESULTS

## D.1  LARGER MODELS

We report the performance of AdaMeZO on larger models to demonstrate the scalability of the optimizer as Table 10 and Table 11.

We also report the hyperparameter settings in Section D.1 for Table 11.

We include training loss and evaluation loss curves in Figure 7 and Figure 8, respectively.

Table 10: Main results on LLaMA-7B over language tasks.

| Task | SST-2 | RTE | CB | BoolQ | WSC | WIC | MultiRC | COPA | ReCoRD | SQuAD | DROP | Avg | Avg (w.o S,D) |
|---|---|---|---|---|---|---|---|---|---|---|---|---|---|
| Type | | | — classification — | | | | | – multiple choice – | | — generation — | | | |
| Zero-shot | 59.7 | 49.8 | 48.2 | 65.0 | 56.7 | 50.6 | 50.5 | 84.0 | 79.9 | 58.6 | 17.5 | 56.4 | 60.4 |
| FO (12 × memory) | 95.0 | 86.0 | 94.1 | 83.1 | 54.5 | 66.2 | 79.3 | 81.2 | 75.4 | 89.2 | 39.7 | 76.7 | 79.4 |
| | (0.5) | (2.2) | (1.7) | (0.5) | (5.8) | (4.9) | (3.0) | (2.2) | (2.3) | (1.0) | (1.0) | – | – |
| MeZO | 85.7 | 54.7 | 58.8 | 68.3 | 58.1 | 56.9 | 60.9 | 82.5 | 78.0 | 71.9 | 30.9 | 64.2 | 67.1 |
| | (1.9) | (0.5) | (3.8) | (1.5) | (2.9) | (1.7) | (2.7) | (1.2) | (1.8) | (4.5) | (1.1) | – | – |
| MeZO-switch | 87.2 | 55.2 | 60.6 | 68.7 | 60.2 | 56.8 | 60.5 | 84.0 | 80.3 | 78.8 | 32.3 | 65.8 | 68.1 |
| | (0.7) | (1.2) | (6.3) | (1.2) | (1.2) | (0.5) | (2.3) | (0.8) | (0.5) | (3.2) | (1.1) | – | – |
| HiZOO | 90.9 | 59.7 | **63.3** | 70.3 | 59.8 | 57.4 | **62.7** | 83.7 | 79.3 | 21.3 | 4.6 | 59.4 | 69.7 |
| | (2.5) | (2.9) | (0.9) | (1.5) | (6.7) | (0.2) | (2.4) | (1.2) | (1.6) | (1.1) | (0.9) | – | – |
| *AdaMeZO* | **91.4** | **61.2** | 62.9 | **70.9** | **60.5** | **57.6** | 62.1 | **84.5** | **80.5** | **84.9** | **36.2** | **68.4** | **70.2** |
| | (2.5) | (2.6) | (1.6) | (2.2) | (2.0) | (1.1) | (2.6) | (3.1) | (0.9) | (0.9) | (2.1) | – | – |

Table 11: Main results on OPT-30B over language tasks, with prefix-learning.

| Task | SST-2 | WSC | WIC | COPA | Avg |
|---|---|---|---|---|---|
| Zero-shot | 56.6 | 38.4 | 50.1 | 81.0 | 56.5 |
| HiZOO | 90.1 | 56.9 | 55.2 | 86.2 | 72.1 |
| | (1.1) | (6.2) | (3.6) | (1.7) | – |
| *AdaMeZO* | **91.1** | **57.6** | **57.3** | **87.0** | **73.2** |
| | (0.6) | (2.4) | (1.5) | (1.4) | – |

Table 12: Hyperparameter settings for Table 11.

| Experiment | Hyperparameters | Values |
|---|---|---|
| HiZOO (prefix) | $B$ | 16 |
| | $\eta$ | $\{5e{-}2, 1e{-}2, 5e{-}3\}$ |
| | $\mu$ | $1e{-}1$ |
| | # prefix tokens | 5 |
| AdaMeZO (prefix) | $B$ | 16 |
| | $\eta$ | $\{7.5e{-}6, 1e{-}5, 2.5e{-}5\}$ |
| | $\mu$ | $1e{-}1$ |
| | # prefix tokens | 5 |

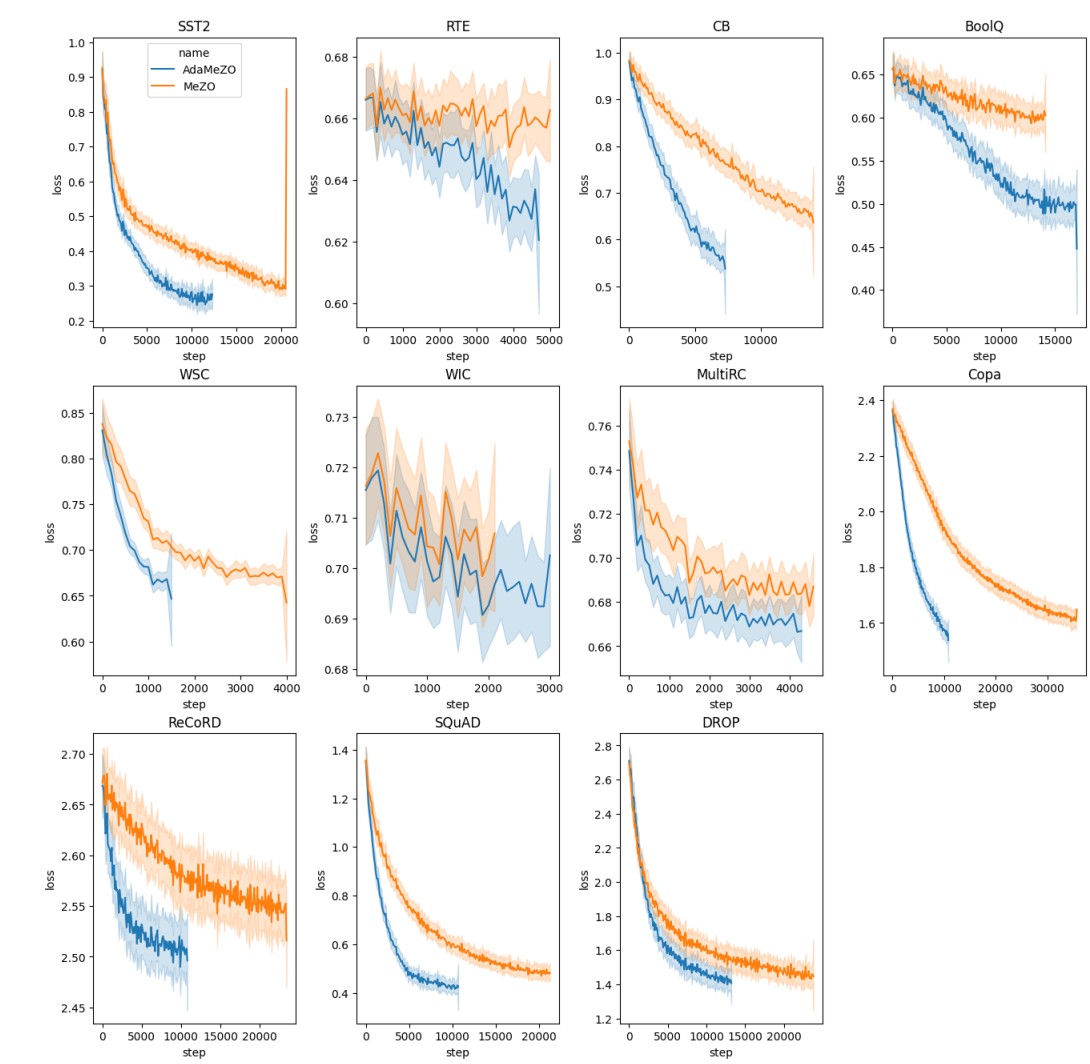

Figure 7: Training loss curve of OPT-13B over language tasks.

Table 13: Hyperparameter settings for HiZOO in Table 3, Table 4, Table 5, and Table 10.

| Experiment | Hyperparameters | Values |
|---|---|---|
| HiZOO | $B$ | 16 |
| | $\eta$ | $\{1e-6, 5e-7, 1e-7\}$ |
| | $\mu$ | $1e-3$ |

## D.2 TIME EFFICIENCY

AdaMeZO incurs longer per-step runtime compared to MeZO, mainly due to a) the additional PRNG calls for past gradient regeneration, and b) the weighted gradient accumulation for moment recovery. We report a runtime profile as Table 14. We can observe that the main contributor to AdaMeZO's additional runtime is the accumulation of regenerated past gradients. Dedicated optimization during the deployment of this accumulation process can speed up the algorithm.

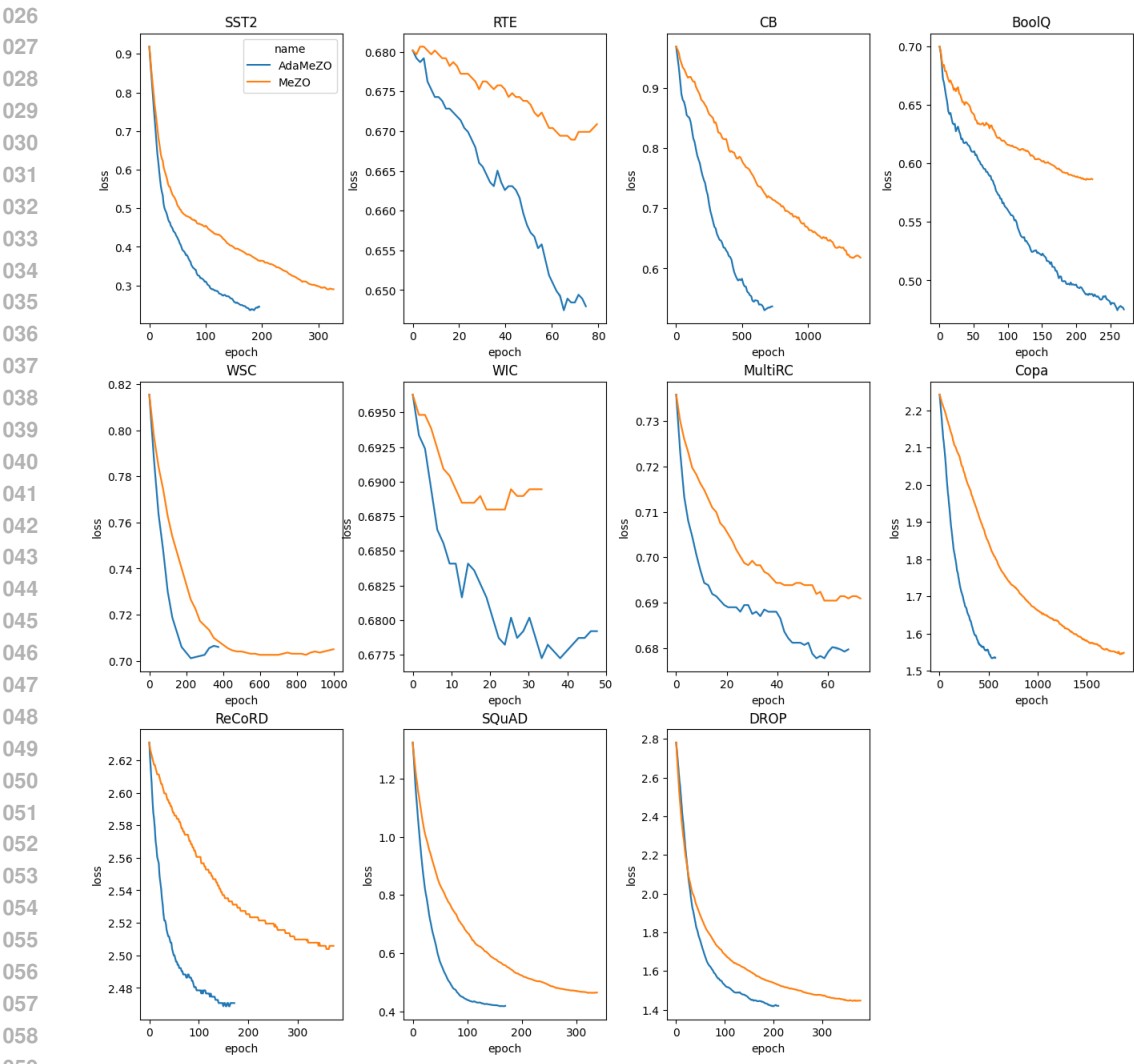

Figure 8: Evaluation loss curve of OPT-13B over language tasks.

Table 14: Runtime profile (sec/step) on standard PyTorch build, measured on OPT-1.3b, batch size=1.

| Optimizer | SST2 | COPA | SQuAD |
|---|---|---|---|
| MeZO | 0.21 | 0.18 | 0.21 |
| HiZOO | 0.23 | 0.24 | 0.25 |
| MeZO + a. | 0.23 | 0.23 | 0.24 |
| MeZO + a. + b. (AdaMeZO) | 0.46 | 0.42 | 0.45 |
| Adam | 0.12 | 0.13 | 0.13 |

## D.3 MEMORY EFFICIENCY

AdaMeZO incurs a small additional memory due to block-wise moment caching. We report a runtime profile as Table 15. We can observe that, compared to optimizers that maintain actual moments, the additional memory cost is significantly reduced.

## D.4 COMPARISON WITH BASELINE THAT MAINTAINS SECOND MOMENT

Table 15: Memory profile (MB) on standard PyTorch build, measured on OPT-1.3b, batch size=1. Measured via `nvidia-smi`.

| Optimizer | SST2 | COPA | SQuAD |
|-----------|------|------|-------|
| MeZO | 5016 | 5058 | 5040 |
| HiZOO | 7532 | 7535 | 7396 |
| AdaMeZO | 5410 | 5452 | 5434 |
| Adam | 11607 | 11529 | 12767 |

We measure the memory cost using the `nvidia-smi` command as reported in Table 6. Notably, it does not accurately reflect the minimal requirement of different baselines due to a series of memory-saving and pre-warming techniques featured by PyTorch. However, the apparent memory cost of zeroth-order methods is still considerably lower than that of first-order methods. Additionally, although the observed runtime of Adam is even shorter than MeZO, under this setting, Adam was unable to train properly due to exploding loss values. One must use gradient accumulation with much longer per-step time or incur much more memory for larger batch sizes.

## E   DETAILED CONVERGENCE ANALYSIS

**Lemma E.1** (Update expectations). *Given Assumption 4.2 to 4.4 and Assumption 4.5, for warm-up steps, it holds that*

$$\mathbb{E}[\boldsymbol{u}_t] = \nabla\mathcal{L}(\boldsymbol{w}_t) + \mathcal{O}(\beta_1 L\eta), \tag{6}$$

$$\mathbb{E}[\|\boldsymbol{u}_t\|_2^2] \leq \frac{\eta^2 L\mathcal{O}(r)}{2}(\|\nabla\mathcal{L}(\boldsymbol{w}_t)\|_2^2 + \sigma^2) + \mathcal{O}(\beta_1 L\eta). \tag{7}$$

*After warm-up steps, it holds that*

$$\mathbb{E}[\boldsymbol{u}_t] = \Sigma_t^{-1}\nabla\mathcal{L}(\boldsymbol{w}_t) + \mathcal{O}(\beta_1 L\eta), \tag{8}$$

$$\mathbb{E}[\|\boldsymbol{u}_t\|_2^2] \leq (2\mathrm{tr}(\Sigma_t^{-1}) + 4s_l^{-1})(\|\nabla\mathcal{L}(\boldsymbol{w}_t)\|_{\Sigma_t^{-1}}^2 + \sigma^2) + \mathcal{O}(\beta_1 L\eta), \tag{9}$$

*where $\sigma$ captures the batch stochasticity in first-order sense, and $\Sigma_t$ is the diagonal matrix with $\boldsymbol{v}_t$ being its diagonal.*

*Proof.* The bounds for the warm-up phase follow Proof of Lemma 2 in Malladi et al. (2023).

After the warm-up case, by the definition of $\boldsymbol{u}_t$, we have

$$\boldsymbol{u}_t = \sum_{i=0}^{h-1}\sum_{j=1}^{n}\frac{\mathcal{L}(\boldsymbol{w}_{t-i} + \mu\boldsymbol{z}_{i,j}, \mathcal{B}_t) - \mathcal{L}(\boldsymbol{w}_{t-i} - \mu\boldsymbol{z}_{i,j}, \mathcal{B}_t)}{2n\mu}\beta_1^i\Sigma_t^{-1}\boldsymbol{z}_{i,j}$$

$$= \sum_{i=0}^{h-1}\sum_{j=1}^{n}\frac{2\beta_1^i\mu\nabla^\top\mathcal{L}(\boldsymbol{w}_{t-i}, \mathcal{B}_t)\boldsymbol{z}_j\Sigma^{-1}\boldsymbol{z}_j + \mathcal{O}(\beta_1 L\eta)}{2n\mu}$$

$$= \sum_{i=0}^{h-1}\frac{\beta_1^i}{n}\sum_{i=1}^{n}\Sigma_t^{-\frac{1}{2}}\boldsymbol{z}_j\boldsymbol{z}_j^\top\Sigma_t^{-\frac{1}{2}}\nabla\mathcal{L}(\boldsymbol{w}_{t-i}, \mathcal{B}_t) + \mathcal{O}(\beta_1 L\eta),$$

$$\mathbb{E}[\boldsymbol{u}_t] = \sum_{i=0}^{h-1}\frac{\beta_1^i}{n}\Sigma_t^{-1}\nabla\mathcal{L}(\boldsymbol{w}_{t-i}) + \mathcal{O}(\beta_1 L\eta)$$

$$= \Sigma_t^{-1}\nabla\mathcal{L}(\boldsymbol{w}_t) + \mathcal{O}(\beta_1 L\eta),$$

where

$$\Sigma_t := \beta_v\sqrt{\sum_{i=0}^{h-1}\beta_2^i\mathrm{diag}\left(\frac{1}{n}\sum_{j=1}^{n}\boldsymbol{g}_{t-i,j}\odot\boldsymbol{g_{t-i,j}}\right)}, \quad \boldsymbol{g}_{t,j} := \boldsymbol{z}_j^\top\nabla\mathcal{L}(\boldsymbol{w}_t, \mathcal{B}_t)\boldsymbol{z}_j,$$

with $\beta_v$ is a normalizing factor connected to $\beta_1$ and $\beta_2$ to cancel out all $\beta_1$ and $\beta_2$ related terms.

Moreover,

$$\mathbb{E}\left[\|\boldsymbol{u}_t\|_2^2\right]$$

$$= \mathbb{E}_{\mathcal{B}_t, \boldsymbol{z}_j}\left[\|\frac{1}{n}\sum_{j=1}^{n}\Sigma_t^{-\frac{1}{2}}\boldsymbol{z}_j\boldsymbol{z}_j^\top\Sigma^{-\frac{1}{2}}\nabla\mathcal{L}(\boldsymbol{w}_t, \mathcal{B}_t) + \mathcal{O}(\beta_1 L\eta)\|_2^2\right]$$

$$\overset{(a)}{\leq} 2\mathbb{E}_{\mathcal{B}_t, \boldsymbol{z}}\left[\|\frac{1}{n}\sum_{i=1}^{n}\Sigma_t^{-\frac{1}{2}}\boldsymbol{z}_j\boldsymbol{z}_j^\top\Sigma_t^{-\frac{1}{2}}\nabla\mathcal{L}(\boldsymbol{w}_t, \mathcal{B}_t)\|_2^2\right] + \mathcal{O}(\beta_1 L\eta)$$

$$\overset{(b)}{\leq} \frac{2}{n}\sum_{j=1}^{n}\mathbb{E}_{\mathcal{B}_t, \boldsymbol{z}_j}\left[\|\Sigma_t^{-\frac{1}{2}}\boldsymbol{z}_j\boldsymbol{z}_j^\top\Sigma^{-\frac{1}{2}}\nabla\mathcal{L}(\boldsymbol{w}_t, \mathcal{B}_t)\|_2^2\right] + \mathcal{O}(\beta_1 L\eta)$$

$$\overset{(c)}{=} 2\mathrm{tr}(\Sigma_t^{-1})\mathbb{E}_{\mathcal{B}_j}\left[\nabla^\top\mathcal{L}(\boldsymbol{w}_t, \mathcal{B}_t)\Sigma_t^{-1}\nabla\mathcal{L}(\boldsymbol{w}_t, \mathcal{B}_t)\right] + 4\mathbb{E}_{\mathcal{B}_j}\left[\nabla^\top\mathcal{L}(\boldsymbol{w}_t)\Sigma_t^{-2}\nabla\mathcal{L}(\boldsymbol{w}_t)\right] + \mathcal{O}(\beta_1 L\eta)$$

$$\overset{(d)}{\leq} (2\mathrm{tr}(\Sigma_t^{-1}) + 4s_l^{-1})\mathbb{E}_{\mathcal{B}_j}\left[\nabla^\top\mathcal{L}(\boldsymbol{w}_t, \mathcal{B}_t)\Sigma_t^{-1}\nabla\mathcal{L}(\boldsymbol{w}_t, \mathcal{B}_t)\right] + \mathcal{O}(\beta_1 L\eta)$$

$$\overset{(e)}{\leq} (2\mathrm{tr}(\Sigma_t^{-1}) + 4s_l^{-1})(\|\nabla\mathcal{L}(\boldsymbol{w}_t)\|_{\Sigma^{-1}}^2 + s_l^{-1}\sigma_t^2) + \mathcal{O}(\beta_1 L\eta).$$

where $(a)$ is by $\|a + b\|_2^2 \le \|a\|_2^2 + \|b\|_2^2 + 2\|ab\|_2 \le 2\|a\|_2^2 + 2\|b\|_2^2$; $(b)$ is by the convexity of the function $\|\cdot\|^2$; $(c)$ is by setting $A = \mathbb{E}_{\mathcal{B}_j}\left[\Sigma_t^{-\frac{1}{2}} \nabla^\top \mathcal{L}(\boldsymbol{w}_t, \mathcal{B}_t) \nabla \mathcal{L}(\boldsymbol{w}_t, \mathcal{B}_t) \Sigma_t^{-\frac{1}{2}}\right]$ and $B = \Sigma_t^{-1}$, then apply Assumption 4.5; $(d)$ is by Assumption 4.3; finally $(e)$ by Assumption 4.2. $\square$

Finally, we establish Theorem 4.7.

*Proof.* Split the full summation into the warm-up phase and the post-warm-up phase as follows.

$$\frac{1}{T}\sum_{t=1}^{T}\|\nabla\mathcal{L}(\boldsymbol{w}_t)\|_2^2 = \underbrace{\frac{1}{T}\sum_{t=1}^{T_w}\|\nabla\mathcal{L}(\boldsymbol{w}_t)\|_2^2}_{\text{warm}-\text{up}} + \frac{1}{T}\sum_{t=T_w+1}^{T}\|\nabla\mathcal{L}(\boldsymbol{w}_t)\|_2^2.$$

Choose

$$\eta \le \min\left\{\frac{1}{s(\operatorname{tr}\Sigma_t^{-1} + 2s_l^{-1})\sqrt{T}}, \frac{1}{L\mathcal{O}(r)\sqrt{T}}, \frac{1}{s\mathbb{E}[\mathcal{L}(\boldsymbol{w}_1)] - \mathbb{E}[\mathcal{L}(\boldsymbol{w}_T)]\sqrt{T}}\right\},$$

Equation (6) and equation 7 with Theorem 4.1 yields

$$\mathbb{E}[\mathcal{L}(\boldsymbol{w}_{t+1})] \le \mathcal{L}(\boldsymbol{w}_t) - \eta\|\nabla\mathcal{L}(\boldsymbol{w}_t)\|_2^2 + \frac{\eta^2 L\mathcal{O}(r)}{2}(\|\nabla\mathcal{L}(\boldsymbol{w}_t)\|_2^2 + \sigma^2) + \mathcal{O}(\beta_1 L\eta)$$

$$\le \mathcal{L}(\boldsymbol{w}_t) - \frac{\eta}{2}\|\nabla\mathcal{L}(\boldsymbol{w}_t)\|_2^2 + \frac{\eta^2 L\sigma^2\mathcal{O}(r)}{2} + \mathcal{O}(\beta_1 L\eta).$$

Equation (8) and equation 9 with Theorem 4.1 yields

$$\mathbb{E}[\mathcal{L}(\boldsymbol{w}_{t+1})] \le \mathcal{L}(\boldsymbol{w}_t) - \eta\|\nabla\mathcal{L}(\boldsymbol{w}_t)\|_{\Sigma_t^{-1}}^2 + L\eta^2(\operatorname{tr}\Sigma_t^{-1} + 2s_l^{-1})\left(\|\nabla\mathcal{L}(\boldsymbol{w}_t)\|_{\Sigma_t^{-1}}^2 + \sigma^2\right) + \mathcal{O}(\beta_1 L\eta)$$

$$\le \mathcal{L}(\boldsymbol{w}_t) - \frac{\eta}{2}\|\nabla\mathcal{L}(\boldsymbol{w}_t)\|_{\Sigma_t^{-1}}^2 + L\eta^2\sigma^2(\operatorname{tr}\Sigma_t^{-1} + 2s_l^{-1}) + \mathcal{O}(\beta_1 L\eta).$$

So, for the warm-up phase,

$$\frac{1}{T}\sum_{t=1}^{T_w}\|\nabla\mathcal{L}(\boldsymbol{w}_t)\|_2^2 \le \frac{2}{\eta T}(\mathcal{L}(\boldsymbol{w}_1) - \mathbb{E}[\mathcal{L}(\boldsymbol{w}_{T_w})]) + \frac{T_w L\eta\sigma^2\mathcal{O}(r)}{T} + \mathcal{O}(\beta_1 L\eta), \qquad (10)$$

and for the post-warm-up phase, Equation (8) and equation 9 with Theorem 4.1 yields

$$\frac{1}{T}\sum_{t=T_w+1}^{T}\|\nabla\mathcal{L}(\boldsymbol{w}_t)\|_2^2$$

$$\le \frac{s_u}{T}\sum_{t=T_w+1}^{T}\|\nabla\mathcal{L}(\boldsymbol{w}_t)\|_{\Sigma_t^{-1}}^2$$

$$\le \frac{2s_u}{\eta T}(\mathbb{E}[\mathcal{L}(\boldsymbol{w}_{T_w+1})] - \mathbb{E}[\mathcal{L}(\boldsymbol{w}_T)]) + \frac{s_u(T - T_w)L\eta\sigma^2(\operatorname{tr}\Sigma_t^{-1} + 2s_l^{-1})}{T} + \mathcal{O}(\beta_1 L\eta). \qquad (11)$$

Take $s = \max\{1, s_u\}$, combine Equation (10) and equation 11,

$$e \le \frac{2s}{\eta T}(\mathbb{E}[\mathcal{L}(\boldsymbol{w}_1)] - \mathbb{E}[\mathcal{L}(\boldsymbol{w}_T)]) + \frac{T_w\eta L\sigma^2\mathcal{O}(r)}{T} + sL\eta\sigma^2(\operatorname{tr}\Sigma_t^{-1} + 2s_l^{-1}) + \mathcal{O}(\beta_1 L\eta)$$

$$\le \frac{L\sigma^2}{\sqrt{T}} + \frac{2}{T\sqrt{T}} + \frac{T_w\sigma^2}{T\sqrt{T}} + \mathcal{O}(\beta_1 L\eta),$$

arriving at the target. $\square$

