# OpenReview forum: "AdaMeZO: Adam-Styled Zeroth-Order Optimizer for LLM Fine-tuning Without Memorizing the Moments"
_ICLR.cc/2026/Conference — Submitted to ICLR 2026_

### Official Review · Reviewer_jDkH · 2025-10-17

**Soundness:** 3
**Presentation:** 3
**Contribution:** 4
**Rating:** 4
**Confidence:** 3

**Summary:**

This paper presents AdaMeZO, an optimization algorithm which combines the benefits of Adam (good choice of weight updates) and MeZO (foregoing backpropagation using a gradient estimator to save memory). Compared to Adam, AdaMeZO uses much less memory since it doesn't store accumulators for the momentum and variance, and compared to MeZO, AdaMeZO applies momentum and variance normalization which improves the quality of the step taken. AdaMeZO uses a clever gradient/EMA estimator that is structured so that it can be discarded and regenerated very quickly whenever needed without taking much memory or time, and this estimator serves as the basis for reducing the memory requirements for training. Experiments show good performance on fine tuning LLMs at the >=billion parameter scale, with improvement over MeZO.

**Strengths:**

- A clever gradient regeneration trick is used to compute EMAs without storing the EMAs themselves.
- Block-wise computation of the Adam update using regenerated gradients from PRNG state caching allows for negligible additional memory cost during the step update computation.
- Experiments show that this method is capable of fine tuning models at the billion parameter scale.

**Weaknesses:**

I would expect that the SPSA gradient estimator to introduce a huge amount of noise into the gradient signal, in such a high dimensional problem. This leaves the good experimental performance unexplained, as I would not ordinarily expect learning to happen at a reasonable rate when the gradients have that much noise in them. There is no explanation offered as to why the noise doesn't slow the learning down to an unreasonable rate. I would increase my score for a good explanation as to how this happens.
- The convergence rate theorem (Theorem 4.1) is not sufficiently described; some variables are not specified. Particularly important is $\sigma^2$, which is the variance of the gradient due to batch stochasticity, serves as a multiplicative constant in the big-O bound on the number of steps needed. Ordinarily I would find this bound to be convincing, but I suspect the variance of the gradient estimates would be massively inflated when we replace backpropagation by SPSA estimation, especially in high dimensional problems. If you can prove that $\sigma^2$ grows sub-linearly with problem dimension with fixed expected gradient magnitude when using SPSA estimation of the gradients, this would offer me some explanation for the above.

**Questions:**

- The expected variance $v_t$ collected from backpropagation style gradients is not going to be the same as the expected $v_t$ as calculated from the formula on line 251, because the inherent noise introduced by SPSA gradient estimation add to the variance estimate, making it larger. Empirically and/or theoretically, what is the relative scale of this additional variance versus the original $v_t$ that we would have gotten from backpropagation gradients? Hopefully it is a small effect.
- Is there any other explanation you can give on how a zero-order optimizer scales efficiently to the billion parameter regime despite the noise? Anything I am not thinking of or misunderstood. Maybe the loss is relatively insensitive to noise in most of the extraneous directions? Maybe somehow the gradient estimator has a low-rank covariance matrix in practical settings?
- A popular alternative for decreasing memory cost in training is to split the batch into sub-batches and perform gradient accumulation. Do we know how AdaMeZO compares for memory cost and wall time consumed during experiment?
- Please include Adam in Tables 10 and 11, since that's what you're hoping to improve over.

---

> ### Author Response · Authors · 2025-11-26
> **Response to reviewer jDkH**
>
> Thanks for finding our work clever in addressing the memory bottleneck and capable of a billion-parameter scale models. We will address your concerns point by point in the following.
>
> **W1**. Thanks for your comment. In fact, the possibility of fine-tuning LLMs with zeroth-order optimizers was first demonstrated by [R1]. Before this work, the consensus was that the quadratic norm of the zeroth-order batch gradient estimation is roughly $d$ times that of the first-order, where $d$ is the problem dimension, forbidding ZO optimizers for LLMs. However, with a low local effective rank assumption works only for fine-tuning (Assumption E.6 in our manuscript, Assumption 2 in [1]), the ratio of the quadratic norm of the zeroth-order batch gradient estimation over the first-order equivalent can be controlled by a dimensionally-independent constant (proof can be found in Page 33 in [R1], inflation rate is denoted as $O(r)$ in our manuscript, and $\sigma$ is the batch stochasticity in first-order sense).
>
> **Q1**. Thanks for your question. We believe that our response to **W1** provides a theoretical sense to some extent. A comprehensive analysis of the empirical analysis of $v$ requires substantial work, especially when considering that $v$ may manifest different patterns across tasks, training stages, model size, and architectures. We aim to conduct an empirical analysis in future research works.
>
> **Q2**. Thanks for your question. Indeed, there is no universal guarantee for zeroth-order optimizers to work well on large problems. However, for dedicated settings like fine-tuning on a well-trained model, zeroth-order optimizers can work well since the intrinsic dimension of the problem is extremely small. It remains prohibitively expensive for zeroth-order optimizers to do from-scratch training.
>
> **Q3**. Thanks for your question. We would like to point out that our work in reducing memory can be applied in parallel with decreasing batch sizes. We can roughly split the memory cost in modern deep learning training as P + A + O + BP, where P stands for weight storage, A for activations, O for optimizers-incurred memory, and BP for backpropagation, respectively. For common LLMs,
>
> - A scales to the batch size.
> - BP is usually ≈10P for first-order methods and 0 for zeroth-order.
> - For SGD, mSGD, and Adam, O = 0, P (full first moment), 2P (full first and second moment), respectively.
>
> For the vanilla MeZO [R1], O=0, BP=0 (using SGD over zeroth-order estimations), and the total cost is P + A. However, if one tries to directly run Adam-style updates over zeroth-order estimations, the total cost jumps to 3P + A, with O = 2P for Adam. This work aims to reduce the total cost from 3P + A to ≈approximately 1.05P + A.
>
> **Q4**. Thanks for your question. We have included the apparent memory cost. We would like to mention that the conclusion by BP, which requires ≈approximately 10 times additional memory, as observed by [R1], is based on carefully controlled conditions and may be less observable in modern builds of excellent tools such as PyTorch. However, we can still observe a significantly higher cost of first-order approaches compared to zeroth-order methods.
>
>
> We hope that our response addresses your concerns. If you have any further questions, please don't hesitate to contact us. Thanks again for the attention you put into reviewing our work.
>
> [R1] Malladi et al. Fine-Tuning Language Models with Just Forward Passes.

---

### Official Review · Reviewer_mcvP · 2025-10-18

**Soundness:** 2
**Presentation:** 1
**Contribution:** 2
**Rating:** 4
**Confidence:** 5

**Summary:**

The paper introduces AdaMeZO, a zeroth-order optimizer for LLM fine-tuning that incorporates Adam-style first and second moment estimation without explicitly storing the moment vectors. The method leverages truncated reconstruction and random state caching to approximate momentum terms efficiently, preserving the low memory footprint of MeZO while improving convergence speed and adaptability.

**Strengths:**

- Addresses a practical and important challenge: efficient fine-tuning of LLMs with limited memory resources.

- The proposed truncation and reconstruction strategy for Adam-style moments is novel and allows moment estimation without prohibitive memory overhead.

- Provides extensive theoretical analysis supporting the convergence properties of the method.

**Weaknesses:**

- The experiments mainly compare with MeZO baseline. It would strengthen the claims to include other baselines to validate the generalization ability of the proposed method.

- The writing feels rushed. Important algorithm, experiment reasults on larger LLMs(13B),  ablation studies on hyperparameters are only in the appendix; maybe these should be brought into the main text for clarity and accessibility.

- The paper currently lacks visualizations of the optimization process, similar to Figure 1. It would be very useful to include loss curves for larger-scale models (e.g., 13B) to demonstrate training dynamics more clearly.

**Questions:**

Seen in weakness

---

> ### Author Response · Authors · 2025-11-26
> **Response to reviewer mcvP**
>
> Thank you for finding our work novel and addressing practical challenges. We will respond to the weaknesses point by point.
>
> **W1**. Thanks for pointing out this issue. In light of this, we attempted to add HiZOO and ZO-AdaMU into our comparison. We successfully reproduced HiZOO and have incorporated it into Tables 3 & 4. However, we found that the official repository of ZO-AdaMU failed to run due to argument errors. Specifically, the function ``self._ema_weight()`` is called in Line 732 of \url{https://github.com/MathIsAll/ZO-AdaMU/blob/main/trainer.py}, but it requires a mandatory argument, as in Line 238. We tried to remedy this by trying several reasonable values, but none of them yielded good results. We have the following observations.
>
> - W1.1. Though HiZOO outperforms in some instances, AdaMeZO has higher average metrics.
>
> - W1.2. There is a huge metric drop in the SQuAD and DROP tasks in LlaMa-3B, HiZOO. We suspected that there were precision problems in HiZOO's EMA diagonal Hessian estimation. We are using FP16 (since we are using NVIDIA Ampere cards) to simulate a realistic memory-constraint setting, whereas the HiZOO paper did not disclose the precision used in the empirical comparison.
>
> **W2**. Thanks for your comment. We relocated some key components to the main text, including the full evaluation of OPT-13B. We will add the complete results of HiZOO on OPT-13B as soon as possible.
>
> **W3**. Thanks for your comment. We have reported the training loss curves and evaluation loss curves in Figures 7 & 8. It is observable that AdaMeZO consistently outperforms MeZO over the full set of fine-tuning tasks.
>
> We hope that our response addresses your concerns. If you have any further questions, please don't hesitate to contact us. Thanks again for the attention you put into reviewing our work.

---

### Official Review · Reviewer_86GX · 2025-10-19

**Soundness:** 2
**Presentation:** 2
**Contribution:** 2
**Rating:** 2
**Confidence:** 4

**Summary:**

This paper introduces **AdaMeZO**, a new member of the Ada-style optimization family. It extends the recently popular zeroth-order optimizer MeZO, focusing on parameter-efficient fine-tuning*of large language models. AdaMeZO proposes a *truncated estimation* of the first- and second-order moments based on sampled gradients, which reduces memory usage by eliminating two full copies of parameter-sized states. Additionally, the paper introduces a *state caching* mechanism to avoid storing the entire second-order momentum before applying the Adam-like update. The authors provide theoretical analysis supporting convergence and empirical results showing that AdaMeZO outperforms MeZO baselines on language model fine-tuning benchmarks.

**Strengths:**

1. The main innovation lies in the *truncated-sum estimation* of the moving average momentum, which is both clever and practical for reducing memory usage.
2. The method achieves *strong fine-tuning performance* compared to MeZO, demonstrating clear empirical advantages.
3. Although unrolling the truncated momentum requires costly $h$ times more gradient samples, the authors mitigate this concern by comparing against an alternate baseline (MeZO-switch), which provides a fairer evaluation.

**Weaknesses:**

1. The review of related work across all three relevant topics (zeroth-order optimization, memory-efficient fine-tuning, and adaptive methods) is limited and misses many recent advances. Important works such as [1][2][3][4][5] should be included to provide sufficient context.
2. Despite the introduction of truncated moment estimation, it remains unclear why the algorithm still needs to compute the full $v_t$ before the update. The paper does not clearly discuss this design choice, and Figure 3 does not effectively illustrate the reason of this challenge.
3. Building on the previous point, the description in Section 3.1.1 is unnecessarily complicated. The key idea seems straightforward: by using different random seeds for each block, one can estimate both $m$ and $v$ without storing an additional copy of state. However, Sections 3.3.1 and 3.3.2 obscure this simplicity with excessive procedural detail.
4. The assumptions underlying Theorem 4.1 are not stated in the main text, making it difficult to interpret the theoretical result. The definitions of $L$, $\sigma$, and $\mu$ should be briefly introduced near the theorem statement for clarity.
5. Similarly, in Appendix E, several notations are undefined. For example, $\Sigma$ is used without being properly introduced.
6. Assumption E4 is particularly strong and makes the theoretical setting overly simplified. The authors should at least provide minimal justification or discuss its reasonableness.
7. In Lemma E7 (line 1000), statements labeled as “E2 to E4” should be marked as *assumptions* rather than *theorems*, likely a typo.
8. The MeZO baseline used for comparison is weak by current standards in zeroth-order optimization. According to Appendix D.5, AdaMeZO struggles to match to baselines that maintain full second-order momentum, which diminishes the claimed advantage.

## Reference

[1] Zhang, Yihua, et al. *Revisiting zeroth-order optimization for memory-efficient LLM fine-tuning: A benchmark.* arXiv:2402.11592 (2024).
[2] Chen, Aochuan, et al. *DeepZero: Scaling up zeroth-order optimization for deep model training.* arXiv:2310.02025 (2023).
[3] Pethick, Thomas, et al. *Training deep learning models with norm-constrained LMOs.* arXiv:2502.07529 (2025).
[4] Defazio, Aaron, et al. *The road less scheduled.* *Advances in Neural Information Processing Systems*, 37 (2024): 9974–10007.
[5] Kunstner, Frederik, et al. *Noise is not the main factor behind the gap between SGD and Adam on transformers, but sign descent might be.* arXiv:2304.13960 (2023).

**Questions:**

1. In Section 5.2, you describe “MeZO-switch—a variant of MeZO where the learning rate is manually adjusted to ensure that its optimization trajectory is longer than that of AdaMeZO.” Could you elaborate on why the *trajectory length* is meaningful in such a very noisy stochastic optimization setting, and how it supports the claim of adaptability rather than underfitting?
2. The computational cost scales linearly with $h$. How does the choice of $h$ influence final performance? Is there an ablation or sensitivity analysis exploring this relationship?

---

I am willing to raise my score if the authors can address the weaknesses and questions above.

---

> ### Author Response · Authors · 2025-11-26
> **Response to reviewer 86GX**
>
> Thanks for finding our work clever and practical. Also, thanks for your considerate review. We will respond to the weaknesses point by point.
>
> **W1**. Thanks for pointing this out. We have added the references to our manuscript. Corresponding parts are highlighted.
>
> **W2**. Thanks for pointing this out. We would like to clarify that MeZO with only the first momentum (or h-MeZO in the manuscript) can be generated in-place with seed control. This is because when in pseudo-codes we write $w_{t+1} = w_{t} - \eta m$, where $m = g_t + \beta_1 g_{t-1} + \beta_2 g_{t-2} + \dots$, the machine instead runs $w \gets w - \beta_1^{\tau} g_{t-\tau}$ for $\tau=0$ to $h$. The machine overwrites $w$ in the memory for $h$ times to complete the optimization step, each time based on the PRNG outputs given by the $\tau$-th seed, within one optimization step, without incurring additional memory.
>
> However, for Adam-style updates, the machine cannot simply overwrite $w \gets w - \beta_1^\tau g_{t-\tau} / v$ similarly, using the PRNG output of the $\tau$-th seed. This is because $v = g_t^2 + \beta_2 g_{t-1}^2 + \beta_2^2 g_{t-2}^2 + \dots$ contains PRNG outputs of all $h$ seeds in the horizon. Attempts to overwrite would destroy the preconditioner, so it has to be prepared before modifying $w$. While a simple solution would be to directly store the full $v$ as in HiZOO, we propose the state control and blockwise update as shown in Figure 4.
>
>
> **W3**. Thanks for the comment. We would like to clarify that we are not using different random seeds for each block. Instead, we use the same set of seeds within each optimization step to reproduce identical $g_t$ values, ensuring consistency with earlier works and allowing $m_t$ and $v_t$ to be correctly recovered. However, to avoid full storage of them, we turn to a blockwise update due to the reasons specified above.
>
> **W4**. Thanks for the comment. We have relocated them to the main text.
>
> **W5**. Thanks for the comment. We have modified the corresponding parts.
>
> **W6**. Thanks for the comment. In light of your comment, we modify the assumption as follows: the difference of the expectation of true gradients of several recent points $\nabla \mathcal{L}(w_{t-i})$ and $\nabla \mathcal{L}(w_t)$ can be controlled by a constant $\mathcal{O}(L \eta)$. This follows from the L-smooth assumption and fits into the intuition that a zero learning rate will result in true gradient stationarity.
>
> **W7**. Thanks for the comment. We have corrected the corresponding parts.
>
> **W8**. Thanks for the comment. We have incorporated HiZOO with a comprehensive task set in Tables 3 & 4, as well as extensive simulations in Table 10.
>
> **Q1**. Thanks for the question. This is based on our observation of the toy examples. From Table 6, we observed that MeZO and AdaMeZO have walked trajectories with comparable (or even higher) lengths. However, in Figure 6, MeZO's trajectory length is obviously shorter than that of AdaMeZO's. This is due to drastic oscillations of MeZO's trajectories without preconditioning. We conjecture that if MeZO cannot outperform AdaMeZO with excessively longer trajectories, then the failure doesn't result from the conservative choices of learning rates, but rather from the inherent limit of MeZO.
>
> **Q2**. Thanks for the question. The optimization computational scales linearly with $h$. We would add an ablation as soon as possible.
>
>
> We hope that our response addresses your concerns. If you have any further questions, please don't hesitate to contact us. Thanks again for the attention you put into reviewing our work.

---

### Official Review · Reviewer_sivg · 2025-11-01

**Soundness:** 2
**Presentation:** 2
**Contribution:** 2
**Rating:** 4
**Confidence:** 4

**Summary:**

This paper proposes AdaMeZO, a memory-efficient zeroth-order optimizer that introduces Adam-style first and second moment estimation without explicitly storing them in memory. The method combines truncated moment approximation and PRNG state caching to mimic Adam-like adaptive preconditioning while retaining MeZO’s forward-only property. The authors present theoretical convergence analysis under a non-convex assumption and empirical evaluations on toy functions and several large language models (RoBERTa-large, OPT-1.3B, and LLaMA-3B).

**Strengths:**

1.This paper proposes AdaMeZO, which achieves training performance comparable to or surpassing MeZO by significantly reducing the number of forward propagations.

2.This paper provides a very detailed discussion of the methodology in Part Three.

**Weaknesses:**

1.The experimental section compares AdaMeZO mainly against MeZO and MeZO-switch. However, several concurrent adaptive zeroth-order optimizers (e.g., HiZOO, Helene, ZO-AdaMU) are only discussed but not empirically compared, even though they are conceptually closest. This weakens the empirical validation and makes it hard to quantify the claimed “70% fewer forward passes” advantage.

2.The experiments stop at RoBERTa-large (350M), OPT-1.3B, and LLaMA-3B, with an appendix extension to 7B and 13B models. While informative, results on larger modern-scale LLMs (≥30B) would be crucial to substantiate the claimed scalability and real-world applicability.

3.The proof in Part Four is too brief due to the excessive discussion in Part Three.

**Questions:**

See Weaknesses.

---

> ### Author Response · Authors · 2025-11-26
> **Response to reviewer sivg**
>
> Thank you for your considerate review and for finding our work competitive in terms of performance and discussion compared to prior wisdom. We will respond to the weaknesses point by point.
>
> **W1**. Thanks for pointing out this issue. In light of this, we attempted to add HiZOO and ZO-AdaMU into our comparison. We successfully reproduced HiZOO and have incorporated it into Tables 3 & 4. However, we found that the official repository of ZO-AdaMU failed to run due to argument errors. Specifically, the function ``self._ema_weight()`` is called in Line 732 of https://github.com/MathIsAll/ZO-AdaMU/blob/main/trainer.py, but it requires a mandatory argument, as in Line 238. We tried to remedy this by trying several reasonable values, but none of them yielded good results. For comparison with HiZOO, we have the following observations.
>
> - W1.1. Though HiZOO outperforms in some instances, AdaMeZO has better average metrics.
>
> - W1.2. There is a huge metric drop in the SQuAD and DROP tasks in LlaMa-3B, HiZOO. We suspected that there were precision problems in HiZOO's EMA diagonal Hessian estimation. We are using FP16 (since we are using NVIDIA Ampere cards) to simulate a realistic memory-constraint setting, whereas the HiZOO paper did not disclose the precision used in the empirical comparison.
>
> **W2**. Thanks for pointing out this issue. Following the HiZOO paper (Table 3), we incorporated OPT-30B into several tasks with prefix fine-tuning, as shown in Table 10 with the settings specified in Table 11. We found that AdaMeZO consistently outperforms HiZOO on tested tasks, including SST2, WSC, WIC, and Copa.
>
> **W3**. Thanks for your comment. We have moved the assumptions to the main text, which was previously in the appendix due to the page limit.
>
> We hope that our response addresses your concerns. If you have any further questions, please don't hesitate to contact us. Thank you again for the attention you put in reviewing our work!

---

### Author Response · Authors · 2025-12-03
**Summary of the Review & Rebuttal**

We would like to extend our gratitude to AC for handling our manuscript and to the reviewers for their time and attention in reviewing. We provide a summary of the reviewers' feedback and our response in this official comment block.

This work proposes AdaMeZO, a low-memory-cost zeroth-order optimizer for LLM fine-tuning with Adam-style moments, without requiring them to be maintained in memory. Reviewers consistently highlighted **practicality and ingenuity** as the **key strengths** of this work:

1. Reviewer ``86GX``:

> The main innovation lies in the  _truncated-sum estimation_  of the moving average momentum, which is both clever and practical for reducing memory usage.

2. Reviewer ``mcvP``:

> Addresses a practical and important challenge: efficient fine-tuning of LLMs with limited memory resources.

3. Reviewer ``jDkH``:

> A clever gradient regeneration trick is used to compute EMAs without storing the EMAs themselves.

The reviewers also have some **similar opinions on the weaknesses** in the original submission, as follows:

**SW1. Lack of stronger baselines;**

- W1 from Reviewer ``sivg``:

> However, several concurrent adaptive zeroth-order optimizers (e.g., HiZOO, Helene, ZO-AdaMU) are only discussed but not empirically compared, even though they are conceptually closest.

- W1 from Reviewer ``mcvp``:

> The experiments mainly compare with MeZO baseline. It would strengthen the claims to include other baselines to validate the generalization ability of the proposed method.

- W8 from Reviewer ``86GX``:

> The MeZO baseline used for comparison is weak by current standards in zeroth-order optimization. According to Appendix D.5, AdaMeZO struggles to match to baselines that maintain full second-order momentum, which diminishes the claimed advantage.

**SW2. Paper organization;**

- W3 from Reviewer ``sivg``:

> The proof in Part Four is too brief due to the excessive discussion in Part Three.

- W4 from Reviewer ``86GX``:

> The assumptions underlying Theorem 4.1 are not stated in the main text, making it difficult to interpret the theoretical result. The definitions of  $L$,  $\sigma$, and $\mu$ should be briefly introduced near the theorem statement for clarity.

- W3 from Reviewer ``mcvP``

> Important algorithm, experiment reasults on larger LLMs(13B), ablation studies on hyperparameters are only in the appendix; maybe these should be brought into the main text for clarity and accessibility.

**Additionally**, Reviewer ``86GX`` and ``jDkH`` commented that they are willing to raise their score if their concerns are solved.

To address the shared weakness of **SW1**, we attempted to reproduce HiZOO and ZO-AdaMU, two strong baseline featuring momentum for convergence improvement. We have incorporated HiZOO into a comprehensive comparison, whereas the official repository of ZO-AdaMU failed to yield satisfactory results due to a missing argument. We include selected results due to length limits.

To address the shared weakness of **SW2**, we reorganized the paper by relocating the assumption section, initially in the Appendix (to satisfy page limits), to the main text, and clarified the meaning of the symbols in the updated PDF file.

In the following, we first present responses to **SW1** and **SW2**, then we discuss **other concerns** raised by the reviewers.

---

> ### Author Response · Authors · 2025-12-03
> **Response to SW1**
>
> We incorporated HiZOO in our evaluation across the complete task set.
>
> ### 1. Main results on OPT-1.3B over language tasks (Table 3)
>
> | Method        | SST-2        | RTE           | CB            | BoolQ         | WSC           | WIC           | MultiRC       | COPA          | ReCoRD        | SQuAD         | DROP          | Avg  | Avg (w.o S,D) |
> |---------------|--------------|---------------|---------------|---------------|---------------|---------------|---------------|---------------|---------------|---------------|---------------|-------|----------------|
> | Zero-shot     | 53.5         | 53.4          | 39.2          | 45.5          | 43.2          | 57.5          | 45.4          | 75.0          | 70.5          | 27.2          | 11.1          | 47.4 | 53.6 |
> | FO (~10x memory usage) | 90.9 (1.2)   | 64.0 (10.7)   | 77.2 (7.9)    | 64.4 (9.3)    | 52.8 (2.0)    | 62.3 (1.9)    | 65.2 (6.0)    | 74.0 (2.9)    | 69.1 (1.2)    | 80.4 (1.5)    | 28.2 (1.7)    | 66.2 | 68.9 |
> | MeZO          | 90.9 (0.3)   | 52.5 (1.5)    | 65.5 (6.9)    | 61.8 (2.1)    | 51.1 (8.4)    | **58.6 (1.4)**| 53.7 (2.2)    | 74.5 (3.6)    | 70.6 (1.0)    | 73.3 (0.2)    | 22.8 (0.6)    | 61.4 | 64.4 |
> | MeZO-switch   | 91.0 (0.6)   | 53.8 (1.6)    | 68.7 (2.3)    | 61.9 (0.6)    | 52.1 (7.6)    | 58.3 (1.6)    | 54.9 (1.5)    | **75.5 (3.6)**| 71.0 (1.2)    | 73.7 (1.2)    | 24.3 (1.3)    | 62.3 | 65.2 |
> | HiZOO         | 90.9 (1.0)   | **54.5 (1.6)**| 63.3 (8.5)    | 62.7 (1.6)    | 49.4 (6.9)    | 58.4 (0.4)    | 55.4 (1.7)    | 74.0 (1.8)    | 70.8 (0.8)    | 74.5 (0.4)    | 24.5 (0.5)    | 61.7 | 64.4 |
> | **AdaMeZO**   | **91.6 (0.3)** | 54.3 (3.1)  | **69.6 (1.4)**| **63.2 (1.6)**| **53.5 (7.8)**| 58.4 (1.6)    | **55.9 (0.7)**| **75.5 (4.0)**| **71.1 (1.3)**| **76.1 (0.7)**| **24.6 (1.0)**| **63.1** | **65.9** |
>
> ### 2. Main results on LLaMA-3B over language tasks (Table 4)
>
> | Method        | SST-2        | RTE           | CB            | BoolQ         | WSC           | WIC            | MultiRC       | COPA          | ReCoRD        | SQuAD         | DROP          | Avg  | Avg (w.o S,D) |
> |---------------|--------------|---------------|---------------|---------------|---------------|----------------|---------------|---------------|---------------|---------------|---------------|-------|----------------|
> | Zero-shot     | 56.0         | 52.7          | 51.6          | 60.9          | 36.5          | 54.3           | 44.8          | 75.0          | 68.2          | 47.3          | 20.8          | 51.6 | 55.5 |
> | FO            | 92.5 (0.7)   | 73.9 (5.4)    | 85.6 (6.9)    | 65.9 (7.3)    | 57.8 (7.6)    | 67.1 (0.7)     | 70.6 (1.8)    | 75.7 (2.6)    | 68.6 (1.0)    | 83.9 (0.3)    | 32.2 (1.8)    | 70.3 | 73.1 |
> | MeZO          | 84.5 (4.9)   | 53.2 (0.7)    | 64.7 (2.6)    | 62.6 (0.7)    | 50.4 (11.3)   | 54.6 (0.3)     | 52.6 (2.5)    | 77.2 (2.0)    | 70.0 (0.4)    | 79.2 (0.9)    | 26.8 (0.5)    | 61.4 | 63.3 |
> | MeZO-switch   | 86.6 (4.5)   | 54.1 (1.5)    | 65.5 (0.9)    | 63.2 (0.3)    | 51.6 (12.2)   | 54.7 (1.0)     | 54.7 (0.6)    | 78.7 (2.2)    | 70.4 (0.6)    | **80.4 (0.9)**| 27.6 (0.6)    | 62.5 | 64.4 |
> | HiZOO         | 92.2 (0.5)   | 54.1 (0.3)    | 65.1 (2.2)    | 63.7 (0.3)    | 52.8 (5.4)    | **54.9 (1.7)** | 56.5 (0.7)    | **82.5 (0.5)**| **71.5 (0.6)**| 18.5 (1.9)    | 6.1 (1.2)     | 56.1 | 65.9 |
> | **AdaMeZO**   | **92.6 (0.5)** | **54.4 (1.5)** | **66.0 (1.4)** | **64.6 (2.6)** | **54.5 (7.5)** | **54.9 (1.6)** | **56.9 (1.0)** | 81.2 (3.2)    | 71.3 (0.9)    | **80.4 (1.8)**| **28.1 (1.1)**| **64.1** | **66.3** |
>
> ### 3. Main results on LLaMA-3B over language tasks (Part of Table 5)
>
> | Method        | BoolQ         | DROP          |
> |---------------|---------------|---------------|
> | Zero-shot     | 59.0          | 14.6          |
> | FO            | 77.1          | 31.3          |
> | MeZO          | 65.5 (3.0)    | 30.6 (1.5)    |
> | MeZO-switch   | 66.2 (3.7)    | 30.5 (0.9)    |
> | HiZOO         | OOM           | OOM           |
> | **AdaMeZO**   | **70.6 (3.6)**| **31.0 (0.9)**|
>
> We adhere to the original hyperparameter settings in HiZOO as outlined in Table 12 of the PDF. Some observations are as follows.
>
> - 1.1. Although HiZOO outperforms in some instances, AdaMeZO has better average metrics, provided that it does not require a large amount of memory to store momentum. The advantage is more pronounced, as HiZOO encounters OOM errors in some tasks (NVIDIA A100 80GB), whereas AdaMeZO successfully terminates with results on par with FO methods.
> - 1.2. There is a significant drop in the SQuAD and DROP tasks for LLaMa models using HiZOO, similar to Llama-7B (see Table 9 in the updated PDF). We suspected that there were precision problems in HiZOO's EMA diagonal Hessian estimation. We are using FP16 (since we are using NVIDIA Ampere cards) to simulate a realistic memory-constraint setting. The HiZOO paper did not disclose the precision used in the empirical comparison.

---

> ### Author Response · Authors · 2025-12-03
> **Response to SW2**
>
> We moved the assumptions to the Appendix to meet page limits, as is done in the HiZOO paper [Zhao et. al, 2024]. We have moved the assumptions parts back to the main text. We have clearly indicated the meanings of the symbols near the main theoretical results. We have also relocated OPT-13B experiments and key ablation on $\beta$ to the main text.
>
> > Zhao at. al, Second-order fine-tuning without pain for llms: A hessian informed zeroth-order optimizer, ICLR'25.

---

> ### Author Response · Authors · 2025-12-03
> **Other weaknesses**
>
> ### Other weaknesses from Reviewer ``sivg``
>
> - Even larger models:
>
> > 2. ...While informative, results on larger modern-scale LLMs (≥30B) would be crucial to substantiate the claimed scalability and real-world applicability.
>
> ★ We report performances on selected tasks with prefix fine-tuning on OPT-30B. We found that AdaMeZO maintains an advantage over HiZOO at the premise of not incurring additional memory to maintain moments.
>
> ### Main results on OPT-30B over language tasks, with prefix-learning (Table 11)
>
> | Method        | SST-2        | WSC           | WIC           | COPA          | Avg  |
> |---------------|--------------|---------------|---------------|---------------|-------|
> | Zero-shot     | 56.6         | 38.4          | 50.1          | 81.0          | 56.5 |
> | HiZOO         | 90.1 (1.1)   | 56.9 (6.2)    | 55.2 (3.6)    | 86.2 (1.7)    | 72.1 |
> | **AdaMeZO**   | **91.1 (0.6)** | **57.6 (2.4)** | **57.3 (1.5)** | **87.0 (1.4)** | **73.2** |
>
> ### Other weaknesses from Reviewer ``86GX``
>
> - References:
>
> > 1. ...Important works such as [1][2][3][4][5] should be included to provide sufficient context.
>
> ★ We have added the references to our manuscript in the related works section. Corresponding parts are highlighted.
>
> - Writing details:
>
> > 2. Despite the introduction of truncated moment estimation, it remains unclear why the algorithm still needs to compute the full $v_t$ before the update. The paper does not clearly discuss this design choice, and Figure 3 does not effectively illustrate the reason of this challenge.
> > 3. Building on the previous point, the description in Section 3.1.1 is unnecessarily complicated. The key idea seems straightforward: by using different random seeds for each block, one can estimate both $m$ and $v$ without storing an additional copy of state. However, Sections 3.3.1 and 3.3.2 obscure this simplicity with excessive procedural detail.
>
> ★ We have made a clearer explanation of the details of our explanation.
>
> - Strong ssumption:
>
> > 6. Assumption E4 is particularly strong and makes the theoretical setting overly simplified. The authors should at least provide minimal justification or discuss its reasonableness.
>
> ★ We have restated our theoretical results under a weaker assumption. In short, we use $L$ in $L$-smooth assumption to control the gradient drift within the horizon, rather than assuming stationary gradients.
>
> - Writings and typos:
>
> > 5. Similarly, in Appendix E, several notations are undefined. For example, $\Sigma$ is used without being properly introduced.
> > 6. In Lemma E7 (line 1000), statements labeled as “E2 to E4” should be marked as  _assumptions_  rather than  _theorems_, likely a typo.
>
> ★ We have fixed the corresponding parts. Thanks to the reviewer's carefulness.
>
> ### Other weaknesses from Reviewer ``mcvP``
>
> - Need more visualizations:
>
> > 3. The paper currently lacks visualizations of the optimization process, similar to Figure 1. It would be very useful to include loss curves for larger-scale models (e.g., 13B) to demonstrate training dynamics more clearly.
>
> ★ We have added loss curves for OPT-13B in Figures 7 & 8.
>
>
> ### Other weaknesses from ``jDkH``
>
> - Billion-level feasibility:
>
> > I would expect that the SPSA gradient estimator to introduce a huge amount of noise into the gradient signal, in such a high dimensional problem. This leaves the good experimental performance unexplained, as I would not ordinarily expect learning to happen at a reasonable rate when the gradients have that much noise in them. There is no explanation offered as to why the noise doesn't slow the learning down to an unreasonable rate. I would increase my score for a good explanation as to how this happens.
>
> ★ We drew from the prior work [Malladi et al., 2023] for support for the feasibility of applying ZO optimizers for billion-level LLM fine-tuning. In short, the previous consensus is that the quadratic norm of the gradient scales to the dimension number $d$, making ZO fine-tuning impossible. However, Malladi et al. introduced a low-effective rank assumption (empirically verified by numerous previous papers), so the inflation rate of the quadratic norm of the gradient estimations is controlled with a constant independent of $d$.
>
> - Sub-batch to reduce memory:
>
> > A popular alternative for decreasing memory cost in training is to split the batch into sub-batches and perform gradient accumulation. Do we know how AdaMeZO compares for memory cost and wall time consumed during experiment?
>
> ★ We clarify that these two methods reduce the memory of different parts. While our work reduces optimizer-incurred memory, batch dividing reduces the memory introduced by activation. They can be applied in parallel, and we have provided a short analysis on this matter.

---

> ### Author Response · Authors · 2025-12-03
> **Thanks again for AC and reviewers for their efforts.**
>
> We would like to extend our gratitude to the reviewers for their thoughtful review. We have improved our work thanks to the high-quality reviews we received. We would also like to extend our appreciation to AC for evaluating our manuscript.

---

### Meta-Review · Area_Chair_PBFF · 2026-01-01

**Summary:**

This paper proposes AdaMeZO, a zeroth-order optimizer that extends MeZO with Adam-style first- and second-moment estimation while avoiding explicit storage of momentum states. The core idea is to reconstruct truncated exponential moving averages via PRNG state caching, aiming to preserve MeZO’s low-memory footprint while improving convergence and adaptability. The paper provides a convergence analysis under non-convex settings and empirical evaluations on several language models, including OPT-1.3B and LLaMA-3B, with additional experiments on larger models added in response to reviewer feedback. Overall, the work targets an important practical problem in memory-efficient LLM fine-tuning and attempts to bridge zeroth-order optimization with adaptive methods.

The paper addresses a relevant and timely challenge, namely improving the efficiency of zeroth-order optimization for large-scale model fine-tuning under strict memory constraints. The truncated moment reconstruction and gradient regeneration mechanism is technically clever and demonstrates a thoughtful engineering effort to avoid storing full-sized optimizer states. Empirically, AdaMeZO consistently outperforms MeZO and its variants across multiple benchmarks while maintaining low memory usage. The authors are responsive to reviewer concerns, adding comparisons with HiZOO, clarifying theoretical assumptions, and improving experimental coverage, which strengthens the revised submission relative to the original.

Despite the revisions, the overall empirical evidence remains somewhat limited. The performance gains are primarily demonstrated relative to MeZO, while comparisons against stronger and more recent adaptive zeroth-order baselines are either incomplete, inconsistent, or rely on implementation-specific issues such as out-of-memory behavior or precision choices. Conceptually, the core idea of reconstructing momentum via reproducible randomness represents an incremental extension rather than a fundamentally new optimization principle. The theoretical analysis still relies on assumptions that are arguably strong, and the explanation of why SPSA-based zeroth-order methods scale effectively to billion-parameter models remains only partially convincing. In addition, the algorithmic presentation is complex relative to the simplicity of the underlying intuition, which negatively affects clarity and accessibility.

In summary, this work presents a technically careful and practically motivated improvement over MeZO, but its contribution appears incremental relative to the rapidly evolving literature on adaptive zeroth-order optimization. While the rebuttal addresses several reviewer concerns, the paper still falls short in terms of demonstrated superiority over strong baselines, theoretical insight into high-dimensional noise behavior, and overall conceptual clarity. As such, I lean toward reject, while encouraging the authors to further strengthen empirical comparisons, simplify the exposition, and more clearly articulate the fundamental reasons behind the method’s effectiveness in large-scale settings.

**Reviewer Concerns:**

The rebuttal addresses several concerns, including clarification of theoretical assumptions and notation, improved presentation, and the addition of comparisons with HiZOO as well as experiments on larger models. The authors also added loss curves and provided further explanation of design choices, which improves clarity relative to the original submission.

However, key concerns remain. The empirical validation against strong adaptive zeroth-order baselines is still limited and not fully convincing, with results relying heavily on MeZO and mixed outcomes against HiZOO. The lack of a solid comparison with other closely related methods remains unresolved. In addition, the theoretical explanation for why SPSA-based zeroth-order methods scale well to billion-parameter models is still incomplete. Overall, while the rebuttal improves the paper, it does not fully address the most critical concerns.

**Reviewer Scores:**

Reviewer sivg would likely maintain a similar score or increase it only slightly. While the added baselines and larger-scale experiments address some of their concerns, the remaining limitations in empirical strength and scalability would probably prevent a clear move into a confident accept.

Reviewer 86GX might modestly increase their score, as several of their concrete concerns about missing baselines, assumptions, and presentation were directly addressed in the rebuttal. However, given their stronger initial skepticism about the strength of the MeZO baseline and overall contribution, a full reversal from reject to accept seems unlikely.

Reviewer mcvP would likely keep their score roughly unchanged or slightly higher. The additional experiments and improved presentation respond to their main concerns, but the overall contribution would still appear marginal relative to acceptance standards.

Reviewer jDkH could increase their score marginally, as the authors provided a partial theoretical explanation for the feasibility of zeroth-order optimization at scale. Nevertheless, since concerns about noise, variance inflation, and scaling behavior remain only partially resolved, a significant score increase is unlikely.

---

### Decision · Program_Chairs · 2026-01-26

Reject